# Global folate deficiency among adolescent girls: A systematic review and meta-analysis

**Mekuriaw Nibret Aweke**[1]*, **Anas Ali Alhur**[2], **Nebebe Demis Baykemagn**[3], **Gebeyehu Lakew**[4], **Bisrat Tewelde Gebretsadkan**[5], **Gebrie Getu Alemu**[6], **Astewil Moges Bazezew**[7], **Amlaku Nigusie Yirsaw**[4], **Wubet Tazeb Wondie**[8], **Berihun Agegn Mengistie**[9]

**1** Department of Human Nutrition, Institute of Public Health, College of Medicine and Health Sciences, University of Gondar, Gondar, Ethiopia, **2** College of Public Health, Imam Abdulrahman Bin Faisal University, Saudi Arabia, **3** Department of Health Informatics, Institute of Public Health, College of Medicine and Health Sciences, University of Gondar, Gondar, Ethiopia, **4** Department of Health Promotion and Health Behavior, Institute of Public Health, College of Medicine and Health Sciences, University of Gondar, Gondar, Ethiopia, **5** Department of Environmental Health and Behavioral Sciences, School of Public Health, College of Medicine and Health Sciences, Mekelle University, Mekelle, Ethiopia, **6** Department of Epidemiology and Biostatistics, Institute of Public Health, College of Medicine and Health Sciences, University of Gondar, Gondar, Ethiopia, **7** Department of Surgical Nursing, School of Nursing, College of Medicine and Health Sciences, University of Gondar, Gondar, Ethiopia, **8** Department of Pediatrics and Child Health Nursing, College of Health Sciences and Referral Hospital, Ambo University, Ambo, Ethiopia, **9** Department of General Midwifery, School of Midwifery, College of Medicine and Health Sciences, University of Gondar, Gondar, Ethiopia

* mekunib@gmail.com

## Abstract

### Introduction

Adolescents have higher nutrient requirements than adults, as this stage accounts for approximately 40% of total adult weight gain, 45% of skeletal mass and about 15% of adult height. Adequate micronutrient intake is particularly essential for adolescent girls to support growth, reproductive health, and cognitive development. Adolescent folate deficiency disrupts hematopoiesis, causing megaloblastic anemia, and hinders growth, cognition, and immune function and becoming a major public health concern.

### Objective

This global systematic review and meta-analysis aims to address the gap in comprehensive evidence on the prevalence of adolescent folate deficiency.

### Methods

This systematic review was conducted and reported according to PRISMA guidelines. We conducted a systematic literature search in PubMed, HINARI, Science Direct, DOAJ, Google, and Google Scholar for studies reporting the prevalence of folate deficiency among adolescent girls up to August 2025. Study quality was assessed using the Newcastle-Ottawa Scale, and evidence certainty was evaluated using

**Data availability statement:** All relevant data are within the paper and its Supporting Information files.

**Funding:** The author(s) received no specific funding for this work.

**Competing interests:** The authors have declared that no competing interests exist.

**Abbreviations:** CI: Confidence Interval; FA: Folic Acid, PRISMA: Preferred Reporting System for Meta-Analysis and Systematic Review; NTDs: Neural Tube Defects; SSA: Sub-Saharan Africa, WHO: World Health Organization.

GRADE. Pooled prevalence estimates with 95% confidence intervals were calculated using a random-effects model (DerSimonian–Laird method). Heterogeneity was assessed with Cochrane Q and I² statistics. Publication bias was evaluated through visual inspection of funnel plots and Egger's regression test, and adjusted estimates were calculated using the trim-and-fill method. Meta-regression analyses were conducted to explore potential sources of heterogeneity. Sensitivity analyses were performed to assess the robustness of pooled estimates.

## Results

The search strategy identified 1,498 records, of which 26 studies met the eligibility criteria and were included in this systematic review and meta-analysis. The pooled global prevalence of folate deficiency among adolescent girls was 26.9%(95% CI: 20.5–33.2), with substantial heterogeneity observed between studies (I² = 99.99%, p < 0.001). On average across included studies, approximately one-quarter of adolescent girls were classified as folate deficient, though the true prevalence varied markedly between settings.. The pooled prevalence of folate deficiency among adolescent girls varied across regions, with the highest rate observed in Africa at 35.5%.

## Conclusion

These findings revealed folate deficiency as a substantial global public health concern among adolescent girls with a disproportionately high burden in low-resource settings. Region-specific strategies are urgently needed to prevent folate deficiency among adolescent girls, particularly in low-resource settings. Implementing targeted nutritional interventions and public health policies is essential to reduce its associated health consequences in this vulnerable population.

## 1. Introduction

Adolescence represents a pivotal stage of human development characterized by rapid growth, profound physiological and hormonal changes with significant shifts in body composition [1]. This stage of life requires optimal nutrition including both macronutrients and micronutrients to support rapid growth, physiological changes, and overall health during adolescence [2].

Adolescents have higher nutrient requirements than adults, as this stage accounts for approximately 40% of total adult weight gain, 45% of skeletal mass and about 15% of adult height [3]. Adequate micronutrient intake is particularly essential for adolescent girls to support growth, reproductive health, and cognitive development [4]. Common essential micronutrients for adolescents girls with particular concern include iron, calcium, zinc, iodine, vitamin D, folate, and vitamin A [5,6].

Folate provides adolescent girls with essential B-vitamin benefits by supporting red and white blood cell production, thereby helping to prevent anemia, and

by aiding DNA and RNA synthesis necessary for rapid growth [7,8]. It also contributes to neurocognitive development, cognitive function, and overall physical performance [9–11]. Folate plays a critical role in the prevention of neural tube defects (NTDs) which are affecting the brain, spine, and spinal cord [12,13]. In adolescent pregnancies, adequate folic acid intake before conception and throughout the first trimester is associated with a reduced risk of spontaneous abortion, preterm delivery, and small-for-gestational-age infants and improved maternal and neonatal outcomes [13–15].

Folate deficiency is a commonly under-recognized nutritional disorder, particularly among adolescents. The World Health Organization (WHO) has established specific biochemical criteria to assess folate status. Red blood cell (RBC) folate reflects long-term folate stores, while serum folate indicates more recent intake [16]. Additionally, elevated homocysteine levels in the absence of vitamin B12 deficiency serve as a functional marker of folate insufficiency, reflecting impaired folate-dependent metabolism [16,17].

Adolescent girls are a nutritionally vulnerable group with exhibiting inadequate dietary intake of several micronutrients including folate [18,19]. Studies conducted in individual countries have reported varying rates of folate deficiency among adolescent girls, with the highest prevalence observed in Sudan (69%) [20]. This deficiency has profound biological and public health implications. Folate deficiency during adolescence impairs hematopoiesis, leading to megaloblastic anemia, and compromises growth, cognitive development, and immune competence [21]. Many adolescent pregnancies are unplanned and inadequate folate status at conception is strongly associated with neural tube defects (NTDs), as well as other obstetric complications such as preterm birth, intrauterine growth restriction, and low birth weight [22,23]. Elevated homocysteine levels, a biomarker of folate deficiency, have also been linked to endothelial dysfunction and increased risk of pregnancy-induced hypertension [23].

Folate deficiency arises from a combination of biological, dietary, genetic, and pharmacological factors that impair folate intake, absorption, metabolism, or increase requirements [17]. These influences can limit intake, reduce absorption, or increase the body's demand for folate. Inadequate dietary intake is one of the primary causes, often due to insufficient consumption of folate-rich foods such as leafy green vegetables and legumes [24]. Folate is also easily destroyed by high cooking temperatures, overcooking, and certain food processing methods [25,26]. Folate status in populations can be assessed using a range of biomarkers. The most common biomarker methods including serum folate levels, red blood cell (RBC) folate concentrations, and urinary folate catabolites such as para-aminobenzoylglutamate and para-acetamidobenzoylglutamate [27–29].

Alternatively, dietary folate intake can be evaluated through self-reported methods, most commonly food frequency questionnaires or quantitative 24-hour dietary recalls [27].

Global efforts to prevent folate deficiency among adolescent girls include supplementation programs, food fortification, and health education initiatives [30]. World Health Organization recommends that adolescent girls and women of reproductive age consume adequate folic acid, particularly before conception and during early pregnancy, to prevent NTDs [31]. Many countries have implemented mandatory fortification of staple foods such as wheat and maize flour, with folic acid, which has been shown to improve folate status and reduce the risk of NTDs [32]. School- and community-based programs, including the WHO-recommended Weekly Iron and Folic Acid Supplementation (WIFAS) for adolescents, aim to increase awareness of folate-rich diets and improve folate and iron status [33].

Despite global efforts such as supplementation programs, food fortification, and health education, folate deficiency remains alarmingly common among adolescent girls, particularly in resource-limited settings. The reported prevalence varies widely across countries, and there is a lack of comprehensive synthesis of the existing evidence. Therefore, conducting a systematic review and meta-analysis is essential to consolidate available data, provide a clear overview of the global burden of folate deficiency among adolescent girls. The objective of this study was to systematically review and quantitatively synthesize the available evidence to estimate the pooled prevalence of folate deficiency among adolescent girls and to inform targeted public health interventions.

 

## 2. Methods

### 2.1 Study protocol and registration

This systematic review and meta-analysis was carried out following the Preferred Reporting Items for Systematic Reviews and Meta-Analyses (PRISMA) guidelines [34]. This review was conducted in strict accordance with the pre-registered PROSPERO protocol (Registration No:CRD420251151715) and no deviations from the protocol were made with respect to eligibility criteria, outcomes, or planned analyses.

### 2.2. Search strategy

Searches were conducted in PubMed, HINARI, Science Direct, DOAJ, Google, and Google scholar to retrieve primary studies reporting the prevalence of folate deficiency among adolescent girls globally. The search for unpublished studies included Google and institutional repositories. Papers were identified using a combination of Medical Subject Headings (MeSH), keywords, and truncations, with the following categories combined using the AND Boolean operator. The keywords used in our searches were: ("folate deficiency" OR "folic acid deficiency" OR "low folate" OR "serum folate" OR "RBC folate" OR "erythrocyte folate" OR "folate insufficiency" OR "folate status") AND (adolescent OR adolescents OR teenager OR teenagers OR youth OR girl OR girls OR female OR "adolescent girls" OR "female adolescents"). All searches were conducted from conception to September 3, 2025, 9:35 EAT by MNA, NDB and GGA. Detailed search strategies are provided in S1 File.

### 2.3. Study selection

A comprehensive collection of articles examining the prevalence and determinants of folate deficiency among adolescent girls was gathered from various sources. There were no restrictions on language or year of publication, and all eligible studies were considered. All studies retrieved through different electronic databases were combined, exported, and managed using EndNote software [35].

### 2.4 Inclusion and exclusion criteria

We included studies published up to September 3, 2025, 11:45 EAT, that reported the prevalence of folate deficiency among adolescent girls globally. Adolescence was defined according to the World Health Organization classification as the age range of 10–19 years, which was used as the eligibility criterion for study inclusion [36]. Titles and abstracts were initially screened against predefined inclusion criteria by AMB, ANY and WTW followed by full-text screening for eligibility. Studies were eligible for inclusion if they met the following criteria: (1) published or unpublished full-text articles up to September 2, 2025; (2) conducted among healthy adolescent girls; and (3) reported the prevalence of folate deficiency.

   Exclusion criteria were: (1) studies that did not report sufficient data to estimate the prevalence of folate deficiency; (2) studies employing qualitative methods, experimental designs, case reports, or case series; (3) studies not conducted among adolescent girls (e.g., adult women, male adolescents, children, or patients and pregnant); (4) studies in which the outcome focused on supplementation rather than folate status; and (5) full text not available. This review was guided by the PECOT framework (Population, Exposure, Comparator, Outcome, Time) to ensure a clear and structured inclusion and exclusion criteria (Table 1).

### 2.5 Assessment of the quality of the individual studies

Three reviewers (MNA, BAM, and TED) independently assessed the quality of the selected studies using a version of the Newcastle-Ottawa Scale modified for cross-sectional study designs. In this review, we used a version of the Newcastle–Ottawa Scale (NOS) modified for cross-sectional studies, assessing study quality across five domains: selection of study groups, sample size, non-respondents, comparability of groups, and ascertainment of the outcome [37]. Any

**Table 1. Criteria for inclusion of studies based on the Population, Exposure, Comparator, Outcome, Time (PECOT) framework.**

| Criteria | Details |
|---|---|
| **Population (P)** | Adolescent girls aged 10–19 years globally. Excludes boys, adult women (>19 years), children (<10 years), pregnant adolescents,adolescent with diagnosis of folate deficiency, and those with gastrointestinal surgeries affecting folate absorption. |
| **Exposure (E)** | Not Applicable |
| **Comparator (C)** | Not applicable |
| **Outcome (O)** | Prevalence of folate deficiency among adolescent girls. |
| **Time (T)** | No restriction on publication year; survey or study period recorded when available. |

discrepancies in evaluation were resolved through discussion until consensus was reached. The tool evaluates key methodological domains, including representatives of the sample, sample size adequacy, non-response rate, validity of measurement tools, comparability of study groups, outcome assessment, and statistical testing [38]. On the basis of these criteria, scores between 0 and 10 were assigned, and studies were categorized into four quality levels: 9–10 as "Very Good," 7–8 as "Good," 5–6 as "Satisfactory," and 0–4 as "Unsatisfactory."

### 2.6 Evidence certainty assessment

In accordance with Cochrane guidelines, we assessed the certainty of evidence on folate deficiency among adolescent girls. The GRADE (Grading of Recommendations, Assessment, Development, and Evaluation) framework was used, considering study design, inconsistency, indirectness, imprecision, and potential publication bias, as described in the GRADE handbook [39]. Certainty of evidence was adapted for proportion estimates and categorized as high, moderate, low, or very low. Findings were summarized in a Summary of Findings (SoF) table generated using the GRADE approach. The assessments were conducted independently by MNA. and GGA., with any disagreements resolved through consensus.

### 2.7 Data extraction

The relevant data were extracted and organized using a standardized table in Microsoft Excel. For each study, information was recorded on the author name, year of publication, study design, country, and region/continent, along with details of the study setting and publication type. Methodological characteristics, including the sampling method, diagnostic criteria, cut-off values, and laboratory techniques, were also documented. Participant information was captured in terms of age range and mean age (S2 File). Data extraction was carried out independently by three reviewers (MNA, TED, and NDB) and any disagreements resolved through discussion until consensus was reached.

### 2.8 Statistical analysis

**2.8.1 Synthesis of results/statistical analysis.** Data analysis was conducted using STATA version 17.0 (StataCorp, College Station, TX, USA). The characteristics of the included studies were summarized in tables and visualized using forest plots. Given the presence of heterogeneity among studies, a random-effects model using the DerSimonian–Laird method was applied to estimate the pooled prevalence of folate deficiency among adolescent girls. This model was chosen because it is more conservative than the fixed-effects model and accounts for variability across studies in meta-analysis. The pooled prevalence, along with the corresponding 95% confidence interval (CI), was reported.

**2.8.2 Sub-group analyses.** Sub-group analyses were performed to explore potential sources of heterogeneity across studies. Stratification was conducted based on factors such as geographical region, sample size, study design, specimen

type for folate assessment, and publication year. This allowed for the assessment of whether the pooled prevalence of folate deficiency varied significantly across different study characteristics.

**2.8.3 Heterogeneity and publication bias.** Statistical heterogeneity was assessed using Cochran's Q test and the I² statistic. Cochran's Q evaluates whether the observed differences in effect sizes are greater than expected by chance. The I² statistic quantifies the proportion of variability due to true heterogeneity rather than random error, with thresholds of 25%, 50%, and 75% typically interpreted as low, moderate, and high heterogeneity, respectively [40]. Publication bias was assessed using funnel plots and Egger's regression test, with a significance level of 0.05 [41].

## 3. Results

### 3.1 Selection and identification of studies

The search identified 1,498 records. After removing duplicates, 1,435 records remained. Titles and abstracts were screened, and 1,357 were excluded. Full texts of 77 articles were assessed for eligibility. Of these, 51 were excluded. Finally, 26 studies with 26,083 adolescent girls were included in the systematic review and meta-analysis on folate deficiency (Fig 1).

### 3.2 Characteristics of included studies

This systematic review included 26 studies with sample sizes ranged from 39 to 13,621 participants with a total of 26,083 adolescent girls [7,20,42–65]. The included studies in this systematic review and meta-analysis were conducted between 1975 and 2024 across multiple regions, including the Americas (USA, Costa Rica) [45,56,57,59,63,64], Africa (Nigeria, Sudan, Ethiopia) [7,20,50], South-East Asia (India, Bangladesh, Sri Lanka, Myanmar) [42,43,46–48,51–53,55,60–62,65], Europe (Turkey) [49,54] and the Eastern Mediterranean (Afghanistan) [44]. Most studies employed a cross-sectional design (n = 23) [7,20,42–48,50–53,55–63,65] while two were retrospective [49,54] and one was longitudinal follow-up study [64]. Regarding to the study setting, the majority being community-based [7,43,45,46,50,51,56,57,59–62,65], school-based [20,42,44,47,48,52,53,55,58,63,64], and two hospital-based [49,54]. Sampling methods included random or probability sampling (n = 9), non-random or convenience sampling (n = 9), and six studies did not specify their sampling method. Folate status was assessed primarily using serum or plasma folate concentration (n = 17), and red blood cell (RBC) folate concentration (n = 9) (table 2). Regarding publication type, 25 studies were peer-reviewed articles, while one government report by the Ethiopian Public Health Institute (EPHI) was classified as gray literature [50]. Of the 26 included studies, 17 (65%) were rated as Satisfactory (scores 5–6), 9 (35%) as Good and none as Very Good [9–10] or Unsatisfactory (0–4) (S3 File).

### 3.3 Global folate defiecency among adolescent girls

The prevalence of folate deficiency among adolescent girls varied widely across countries, from 1.4% in India to 69% in Sudan. Significant heterogeneity was observed across the included studies therefore, a random-effects model was used for all meta-analyses.. Based on the random-effects model, the pooled prevalence of folate deficiency among adolescent girls was 26.9% (95% CI: 20.5–33.2), with substantial heterogeneity observed between studies (I² = 99.99%, p < 0.001). Given the substantial heterogeneity observed between studies, this estimate should be interpreted as an average across diverse contexts rather than as a precise global prevalence. The pooled prevalence was illustrated using a forest plot (Fig 2).

**Subgroup analysis.** To explore potential sources of heterogeneity in folate deficiency among adolescent girls, subgroup analyses were conducted by WHO region, study design, study setting, type of blood sample, and sample size of each study. By WHO region, the prevalence was highest in the African Region at 35.46% (95% CI: 7.9%, 63.2%; I² = 99.99). The lowest pooled proportion was observed in a single study from the Eastern Mediterranean Region which reported 7.4% (Fig. 3).

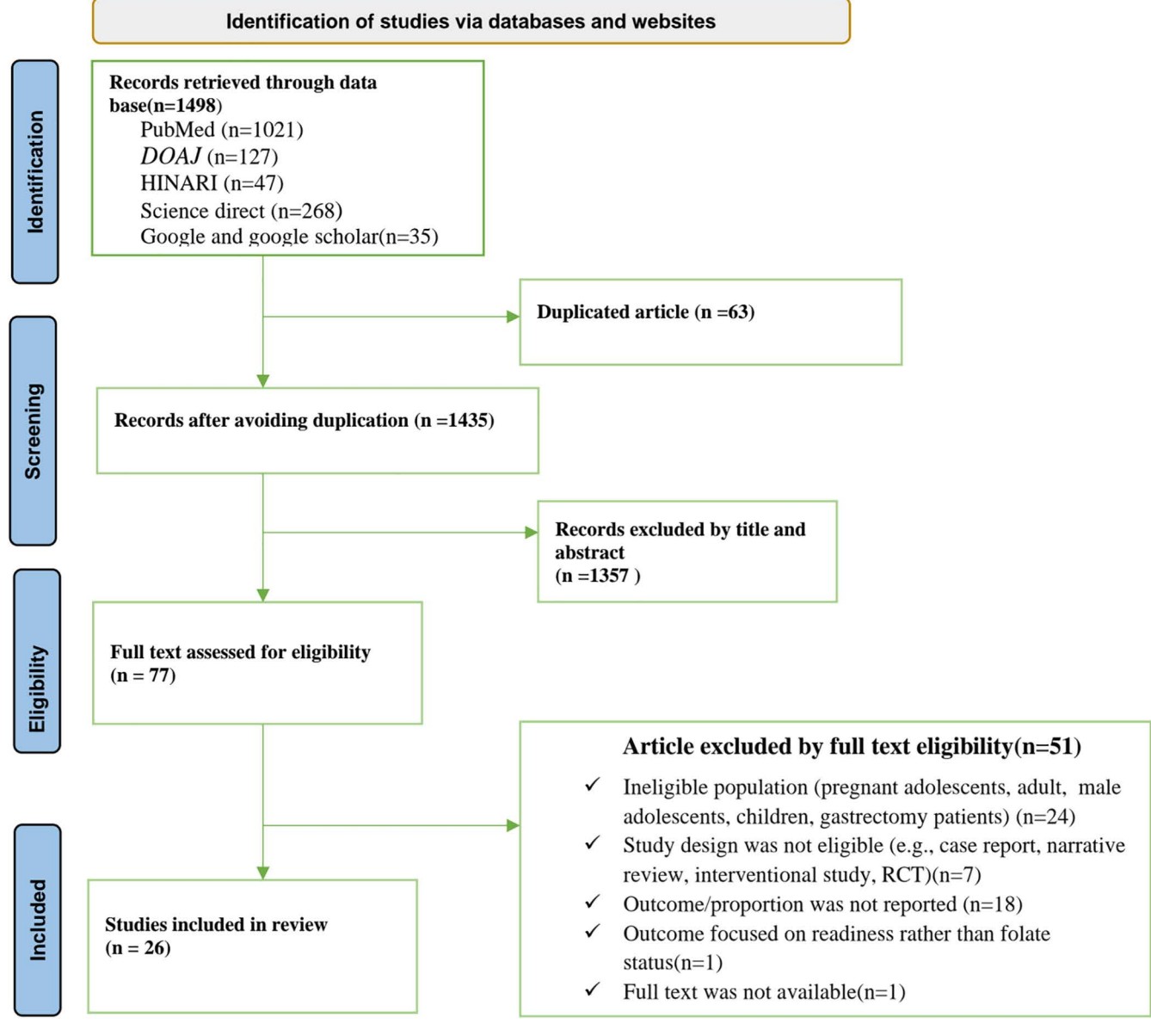

**Fig 1. PRISMA flow diagram of article selection for systematic review and meta-analysis of folate deficiency among adolescent girls 2025.**

To assess temporal variation in folate deficiency among adolescent girls, studies were stratified by publication year into three periods: 1975–2007, 2008–2020, and 2021–2024. The highest pooled prevalence was reported between 2006 and 2016 at 34.4% (95% CI: 22.5%, 46.2%; I²= 99.98%). The lowest prevalence was observed in recent published studies from 2018–2024 at 21.9% (95% CI: 11.2%, 32.5%; I²= 99.97%) (Fig 4).

Studies were stratified by sample size into two groups: < 500 participants and ≥ 500 participants. The pooled prevalence among studies with <500 participants was 27.4% (95% CI: 20.6, 34.2%; I²= 99.98%). For studies with ≥500 participants, the pooled prevalence was 26.0% (95% CI: 14.9%, 37.0%; I²= 100.00%) (Fig 5).

**Table 2. Characteristics of the included studies and their proportion of folate deficiency among adolescent girls, 2025.**

| S.no | Author | Year | Study design | Country | Study setting | Type of sample taken | Mean of age | Sample Size | Preva-lence(%) | Qual-ity |
|------|--------|------|--------------|---------|---------------|----------------------|-------------|-------------|----------------|----------|
| 1 | Daniel Jr WA et al.[45] | 1975 | Cross-sctional | USA | Community-based | Plasma folate concentration | Not reported | 169 | 4.7 | Satis-factory |
| 2 | Liebman M. [56] | 1985 | Cross-sectional | USA | Community based | RBC folate concentration | Not reported | 91 | 32 | Satis-factory |
| 3 | Reiter LA et al. [59] | 1987 | Cross-sectional | USA | Community-based | Serum folate concentration | Not reported | 39 | 3 | Satis-factory |
| 4 | Clark et al.[64] | 1987 | Longitudinal study | USA | Community-based | RBC folate concentration | Not specified | 103 | 47.6 | Satis-factory |
| 5 | Tsui JC et al.[63] | 1990 | Cross-sectional | USA | School-based | Serum folate concentration | Not reported | 164 | 40 | Satis-factory |
| 6 | VanderJagt DJ et al.[7] | 2000 | Cross-sectional | Nigeria | Community-based | Serum folate concentration | Not reported | 162 | 2.4 | Satis-factory |
| 7 | Monge-Rojas et al.[57] | 2005 | Cross-sectional | Costa Rica | Community based | Serum folate concentration | 12.0±1.3 yrs | 49 | 53.3 | Satis-factory |
| 8 | Öner et al. [58] | 2006 | Cross-sectional | Turkey | School-based | Serum folate concentration | Not reported | 704 | 16.3 | Good |
| 9 | Thoradeniya T et al.[62] | 2006 | Cross-sectional | Sri Lanka | Community-based | Serum folate concentration | 17±1·32 yrs | 277 | 45.1 | Good |
| 10 | Ahmed F et al. [42] | 2008 | Cross-sectional | Bangla-desh | School-based | RBC folic acid concentration | Not reported | 310 | 25 | Satis-factory |
| 11 | Abdelrahim II et al.[20] | 2009 | Cross-sectional | Sudan | School-based | Serum folate concentration | 13.9±1.2 | 186 | 69 | Satis-factory |
| 12 | De Lanerolle-Dias et al.[46] | 2012 | Cross-sectional | Sri Lanka | Community-based | Serum folate concentration | 17.5±1.2 years | 600 | 28 | Good |
| 13 | Bansal PG et al. [51] | 2015 | Cross-sectional | India | Community-based | Serum folate concentration | 13.5±2.1 years | 794 | 5 | Satis-factory |
| 14 | Jani R. et al.[53] | 2015 | Cross-sectional | India | School-based | RBC folate concentration | 14.0±1.3 yrs | 224 | 47.3 | Satis-factory |
| 15 | Htet MK et al. [52] | 2016 | Cross-sectional | Myanmar | School-based | Serum folate concentration | 15.9±1.2yrs | 389 | 39.3 | Satis-factory |
| 16 | Ercan S et al. [49] | 2018 | Retrospective | Turkey | Hospital based | Serum folate concentration | Not reported | 55 | 1.8 | Satis-factory |
| 17 | Kumar KJ et al. [55] | 2020 | Cross-sectional | India | School-based | Serum folate concentration | Not reported | 100 | 19 | Satis-factory |
| 18 | Karakaş NM et al.[54] | 2021 | Retrospective | Turkey | Hospital based | Serum folate concentration | 13.84 6±0.9 | 624 | 36.2 | Good |
| 19 | Saxena R et al. [60] | 2021 | Cross-sectional | Bangla-desh | Community-based | RBC folate concentration | 14.6±0.7 yrs | 579 | 17 | Good |
| 20 | Awasthi S et al. [43] | 2022 | Cross-sectional | India | Community-based | Serum folate concentration | Not reported | 1193 | 53.3 | Good |
| 21 | Demuyakor ME et al.[47] | 2023 | Cross-sectional | Bangla-desh | School-based | Serum folate concentration | Not reported | 2159 | 3 | Satis-factory |
| 22 | EPHI.[50] | 2023 | Cross-sectional | Ethiopia | Community-based | Serum folate concentration | Not reported | 2399 | 35 | Good |
| 23 | Shalini T et al. [61] | 2023 | Cross-sectional | India | Community-based | RBC folate concentration | 14.3 yrs±0.1 yrs | 13621 | 32.9 | Good |
| 24 | Basiry M et al. [44] | 2024 | Cross-sectional | Afghani-stan | School-based | Serum folate concentration | Not reported | 380 | 7.4 | Satis-factory |
| 25 | Dhurde VS et al. [48] | 2024 | Cross-sectional | India | School-based | Serum folate concentration | Not reported | 212 | 1.4 | Satis-factory |

*(Continued)*

**Table 2.** (Continued)

| S.no | Author | Year | Study design | Country | Study setting | Type of sample taken | Mean of age | Sample Size | Preva-lence(%) | Qual-ity |
|------|--------|------|--------------|---------|---------------|---------------------|-------------|-------------|----------------|----------|
| 26 | Doshi et al.[65] | 2024 | Cross-sectional | India | School-based | RBC folate concentration | 12.3 years | 500 | 33.3 | Good |

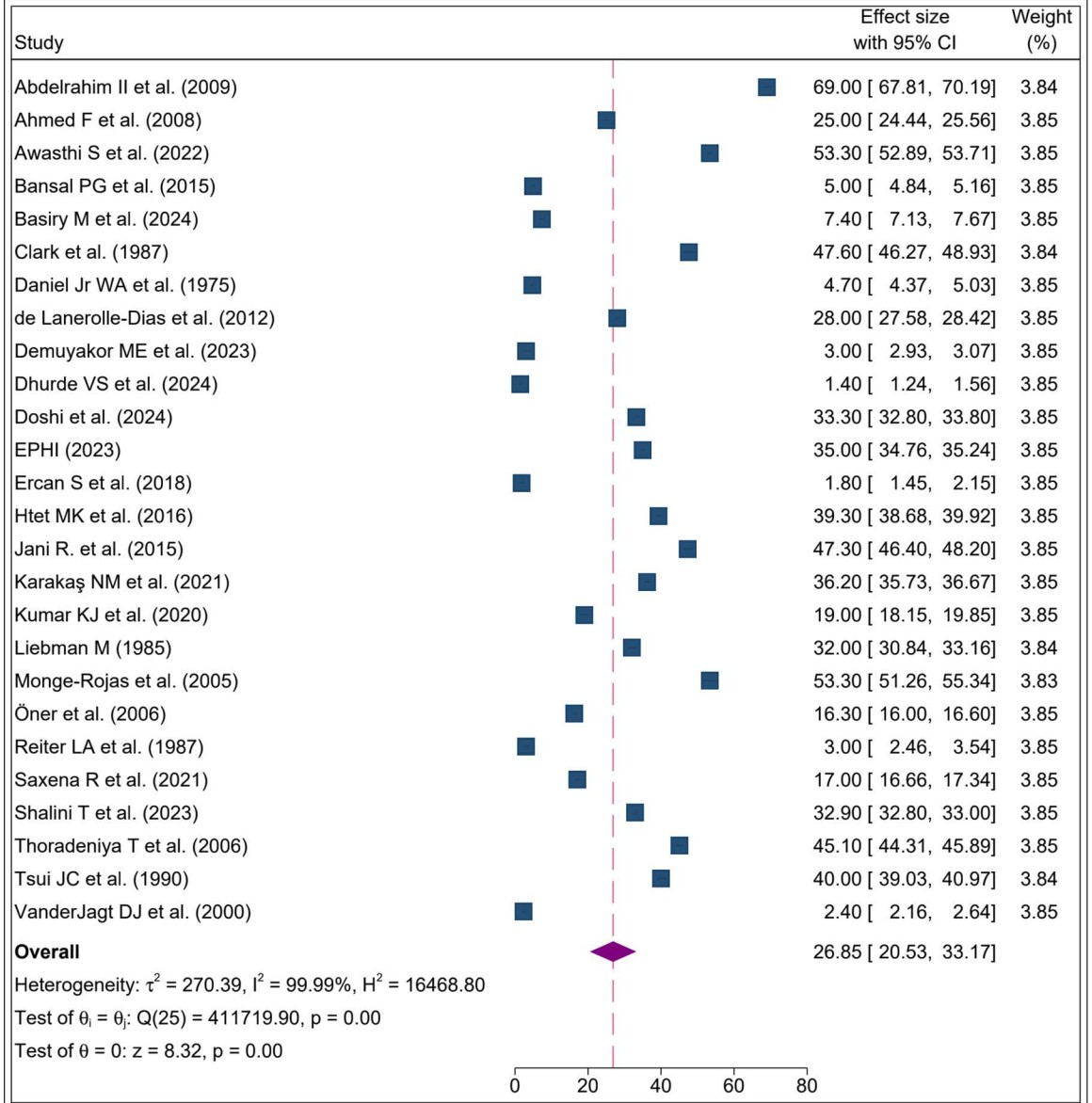

Prevalence of Folate Defiecency

Random-effects DerSimonian–Laird model

**Fig 2. Forset plot for the proportion of folate defiecency among adolecent girls.**

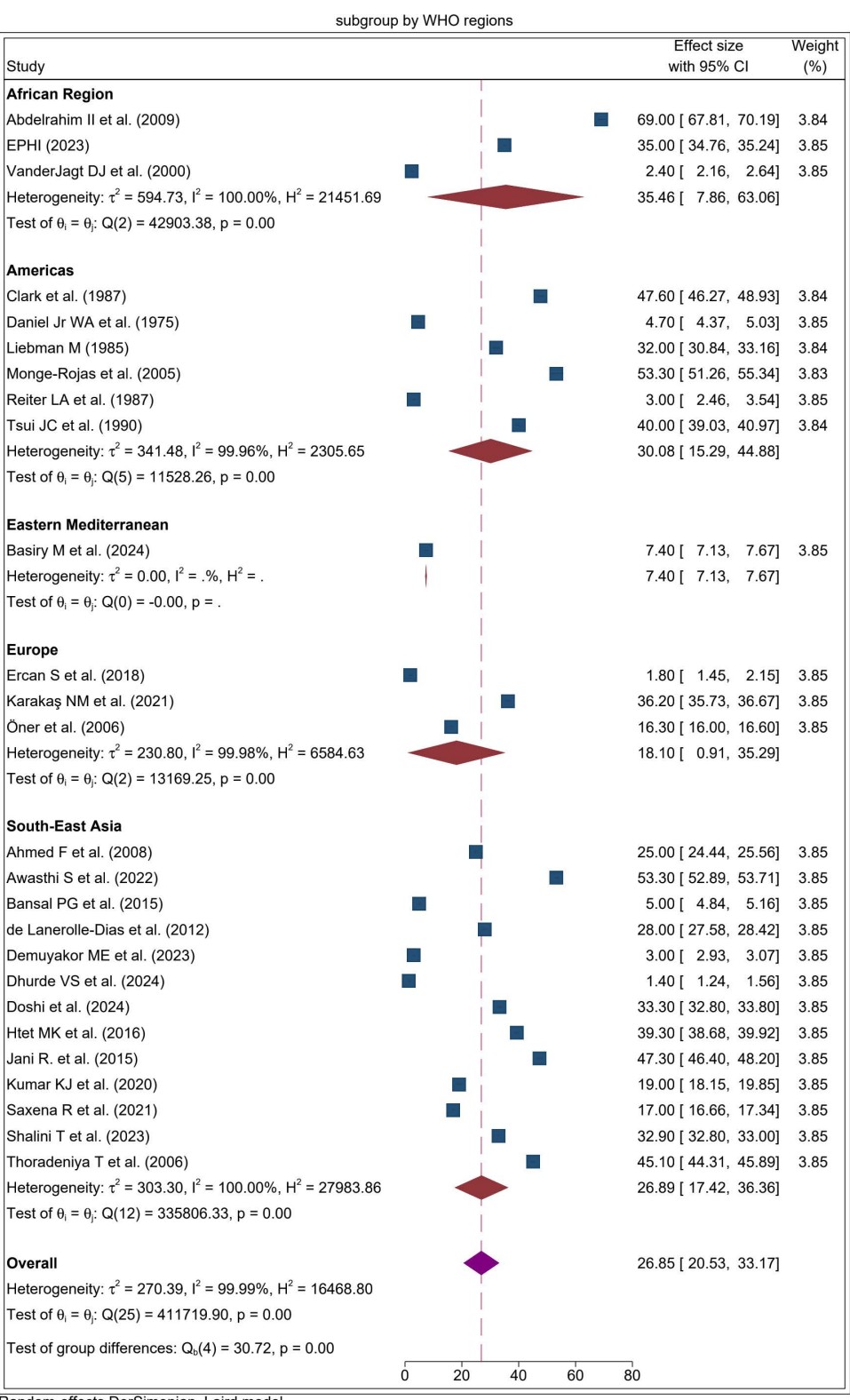

**Fig 3. Forest plot result of subgroup analysis by WHO region for the proportion of folate defiecency among adolecent girls, 2025.**

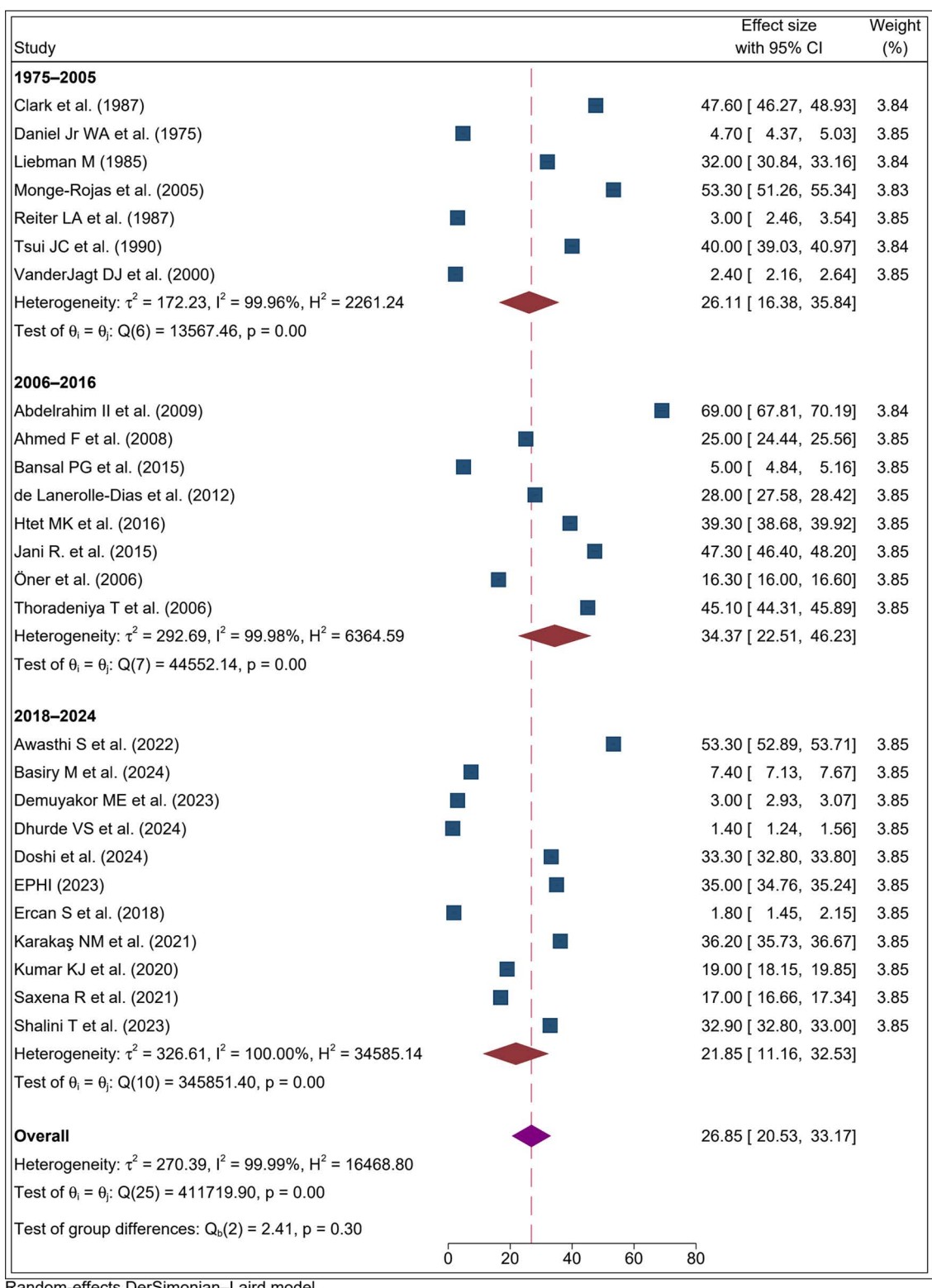

**Fig 4. Forest plot result of subgroup analysis by publication year for the proportion of folate defiecency among adolecent girls, 2025.**

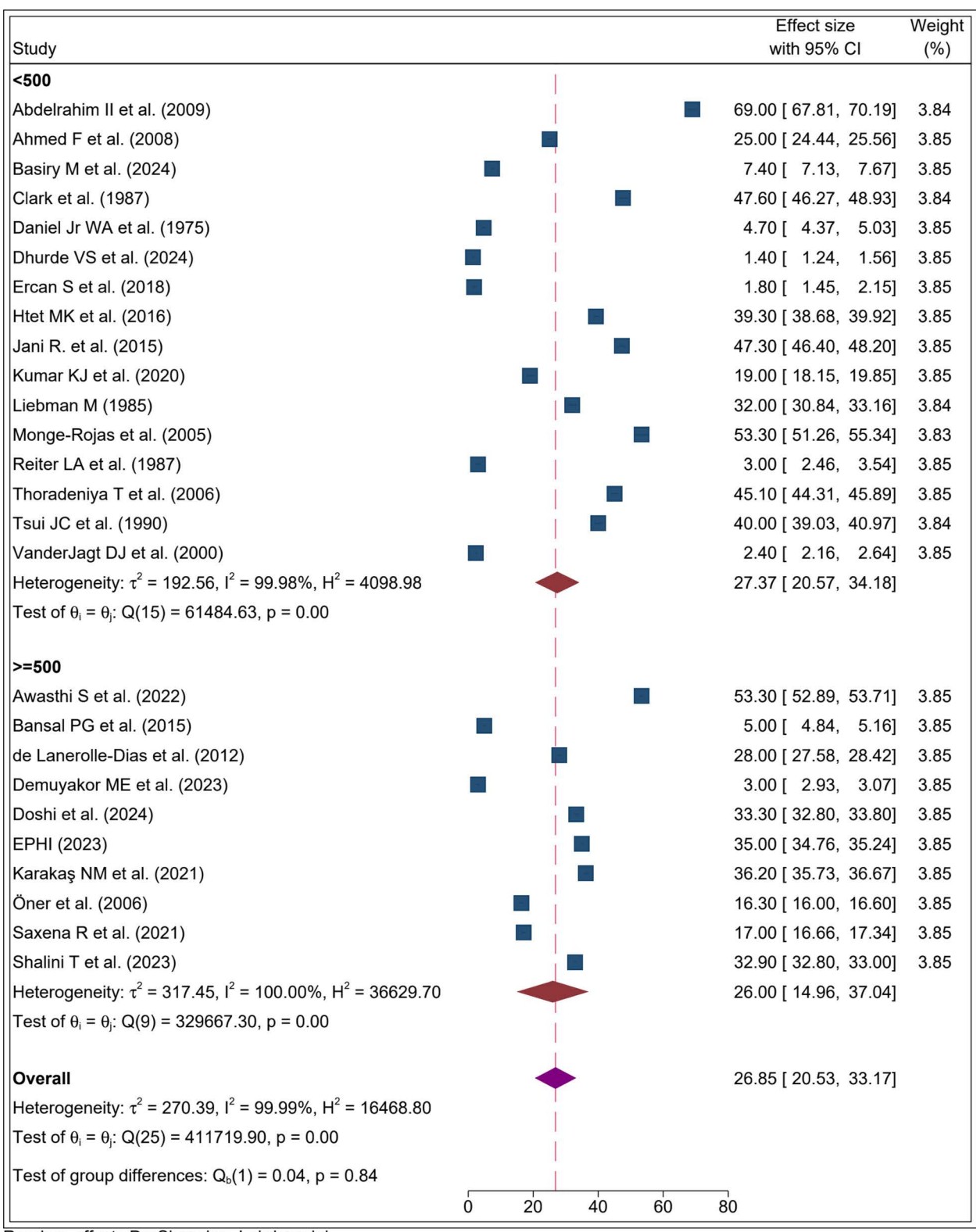

| Study | Effect size with 95% CI | Weight (%) |
|---|---|---|
| **<500** | | |
| Abdelrahim II et al. (2009) | 69.00 [ 67.81, 70.19] | 3.84 |
| Ahmed F et al. (2008) | 25.00 [ 24.44, 25.56] | 3.85 |
| Basiry M et al. (2024) | 7.40 [ 7.13, 7.67] | 3.85 |
| Clark et al. (1987) | 47.60 [ 46.27, 48.93] | 3.84 |
| Daniel Jr WA et al. (1975) | 4.70 [ 4.37, 5.03] | 3.85 |
| Dhurde VS et al. (2024) | 1.40 [ 1.24, 1.56] | 3.85 |
| Ercan S et al. (2018) | 1.80 [ 1.45, 2.15] | 3.85 |
| Htet MK et al. (2016) | 39.30 [ 38.68, 39.92] | 3.85 |
| Jani R. et al. (2015) | 47.30 [ 46.40, 48.20] | 3.85 |
| Kumar KJ et al. (2020) | 19.00 [ 18.15, 19.85] | 3.85 |
| Liebman M (1985) | 32.00 [ 30.84, 33.16] | 3.84 |
| Monge-Rojas et al. (2005) | 53.30 [ 51.26, 55.34] | 3.83 |
| Reiter LA et al. (1987) | 3.00 [ 2.46, 3.54] | 3.85 |
| Thoradeniya T et al. (2006) | 45.10 [ 44.31, 45.89] | 3.85 |
| Tsui JC et al. (1990) | 40.00 [ 39.03, 40.97] | 3.84 |
| VanderJagt DJ et al. (2000) | 2.40 [ 2.16, 2.64] | 3.85 |
| Heterogeneity: $\tau^2$ = 192.56, $I^2$ = 99.98%, $H^2$ = 4098.98 | 27.37 [ 20.57, 34.18] | |
| Test of $\theta_i = \theta_j$: Q(15) = 61484.63, p = 0.00 | | |
| | | |
| **>=500** | | |
| Awasthi S et al. (2022) | 53.30 [ 52.89, 53.71] | 3.85 |
| Bansal PG et al. (2015) | 5.00 [ 4.84, 5.16] | 3.85 |
| de Lanerolle-Dias et al. (2012) | 28.00 [ 27.58, 28.42] | 3.85 |
| Demuyakor ME et al. (2023) | 3.00 [ 2.93, 3.07] | 3.85 |
| Doshi et al. (2024) | 33.30 [ 32.80, 33.80] | 3.85 |
| EPHI (2023) | 35.00 [ 34.76, 35.24] | 3.85 |
| Karakaş NM et al. (2021) | 36.20 [ 35.73, 36.67] | 3.85 |
| Öner et al. (2006) | 16.30 [ 16.00, 16.60] | 3.85 |
| Saxena R et al. (2021) | 17.00 [ 16.66, 17.34] | 3.85 |
| Shalini T et al. (2023) | 32.90 [ 32.80, 33.00] | 3.85 |
| Heterogeneity: $\tau^2$ = 317.45, $I^2$ = 100.00%, $H^2$ = 36629.70 | 26.00 [ 14.96, 37.04] | |
| Test of $\theta_i = \theta_j$: Q(9) = 329667.30, p = 0.00 | | |
| | | |
| **Overall** | 26.85 [ 20.53, 33.17] | |
| Heterogeneity: $\tau^2$ = 270.39, $I^2$ = 99.99%, $H^2$ = 16468.80 | | |
| Test of $\theta_i = \theta_j$: Q(25) = 411719.90, p = 0.00 | | |
| | | |
| Test of group differences: $Q_b$(1) = 0.04, p = 0.84 | | |

Random-effects DerSimonian–Laird model

**Fig 5. Forest plot result of subgroup analysis by sample size for the proportion of folate defiecency among adolecent girls, 2025.**

Finally, Studies were grouped by biological specimen type: red blood cell (RBC) folate and serum/plasma folate concentration. The pooled prevalence among studies using RBC folate concentration was 33.6% (95% CI: 26.9%, 40.2%; $I^2 = 99.94\%$). Studies measuring serum/plasma folate concentration reported a pooled prevalence of 24.4% (95% CI: 18.3%, 30.4%; $I^2 = 99.99\%$) (Fig 6).

### 3.4 Diagnostic techniques used for folate measurement

Folate deficiency among adolescent girls was assessed using a variety of biochemical methods across studies. Serum or plasma folate concentrations were most commonly measured. In meta-analyses evaluating folate deficiency, individual studies have used a range of cut-off values for both red blood cell (RBC) and serum folate concentrations. RBC folate thresholds typically ranged from 140 to 151 ng/mL (approximately 317–340 nmol/L), depending on the assay method employed. Serum folate cut-offs varied more widely, most commonly between 2 and 3 ng/mL (≈4.5–6.8 nmol/L). However, some studies used slightly higher thresholds, such as 4 ng/mL or 6.8–7 nmol/L often based on microbiological or immunoassay techniques (Table 3).

### 3.5 Publication bias across studies

The presence of publication bias was assessed both visually using the symmetry of funnel plots, and statistically, through the Egger's test. Visual inspection of the funnel plot showed that studies were widely scattered due to heterogeneity but were approximately evenly distributed on either side of the pooled effect estimate. This result is suggesting the absence of publication bias (Fig 7).

Similarly, the Egger's regression test indicated no statistically significant evidence of publication bias at the 5% level of significance (Table 4).

Trim-and-fill analysis was conducted to estimate the effect sizes of potentially missing studies and adjust for publication bias. Trim-and-fill analysis imputed 8 potentially missing studies. The funnel plot after imputation appeared symmetric, supporting this conclusion (Fig 8).

### 3.6 Sensitivity analysis

Sensitivity analysis was conducted by excluding each study one at a time to examine its impact on the pooled prevalence of folate deficiency among adolescent girls. The recalculated prevalence ranged from 25.16% (Abdelrahim II et al., 2009) to 27.89% (Dhurde VS et al., 2024), with the overall combined prevalence remaining stable at 26.9% (95% CI: 20.5–33.2). These results indicate that no single study significantly influenced the overall estimate (Fig 9).

### 3.7 Heterogeneity

We assessed heterogeneity in our review using both fixed-effects and random-effects models. This approach allowed us to capture variation within and between studies. To understand the sources of heterogeneity, we carried out sensitivity checks, subgroup analyses, and meta-regression. Despite these efforts, a high level of heterogeneity was still observed. The meta regression results showed that neither publication year nor sample size had a significant effect (Table 5).

### 3.8 Evidence certainty

Overall, the certainty of the evidence regarding the pooled proportion estimates assessed by the GRADE approach was low (Table 6). Although most individual studies were of high methodological quality, the overall certainty of evidence was rated as low due to substantial heterogeneity and indirectness, as high study-level quality does not fully compensate for inconsistency and indirectness across the body of evidence. The directness of the evidence was rated as direct, and the precision of the proportion estimate was satisfactory. However, evidence of publication bias contributed to downgrading the certainty.

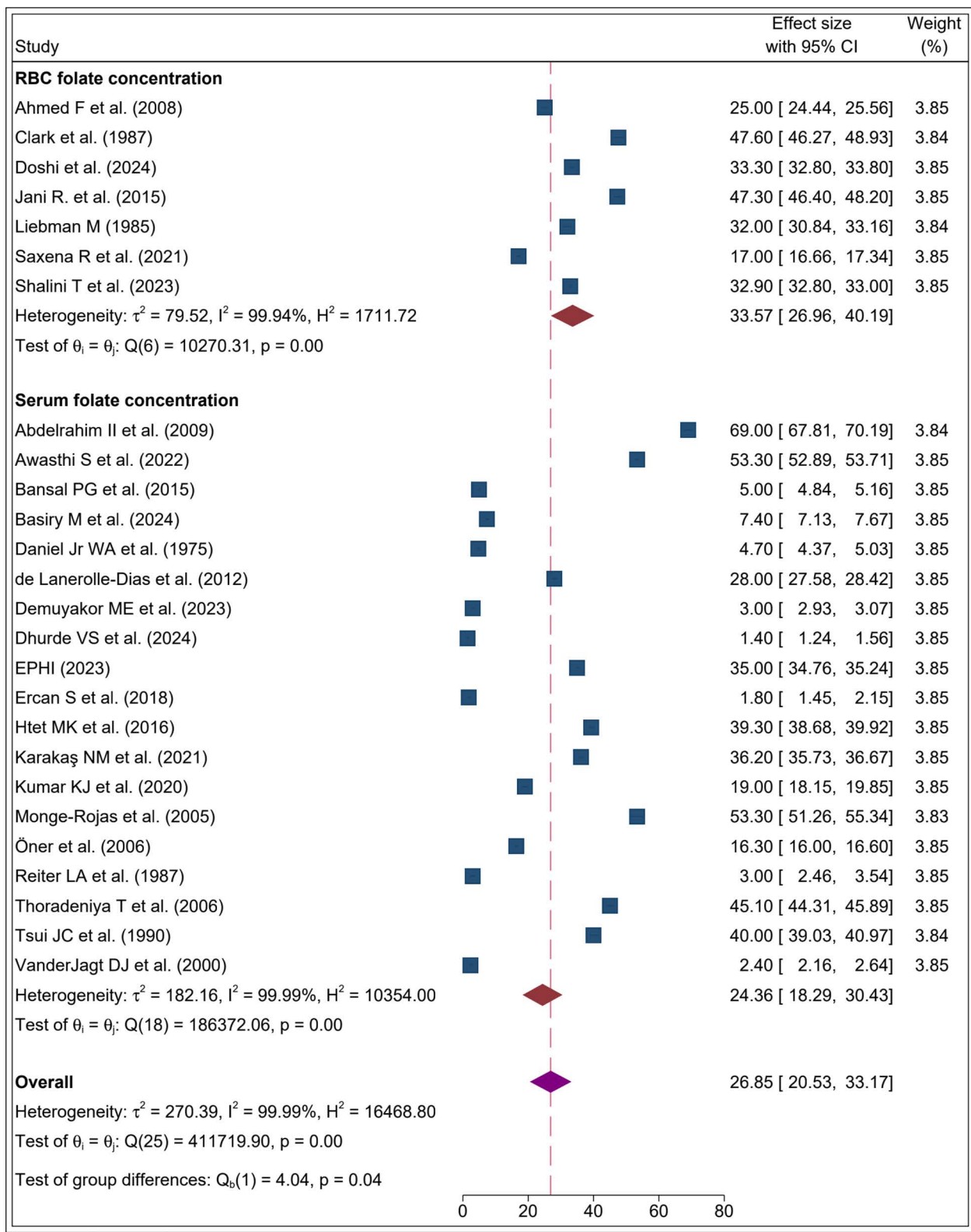

**Fig 6. Forest plot result of subgroup analysis by type of biological sample for the proportion of folate deficiency among adolecent girls, 2025.**

**Table 3. Study-specific cut-off values for red blood cell and serum folate concentrations for the study the proportion of folate defiecency among adolecent girls, 2025.**

| S.no | Author | Diagnostic criteria | Cut-off values | Laboratory technique |
|------|--------|---------------------|----------------|----------------------|
| 1 | Abdelrahim II et al. | Serum folate concentration | <3 ng/ml | Immunofluorescent assay (IMMULITE, SIEMENS) |
| 2 | Ahmed F et al. | RBC folate concentration | <317 nmol/l | Red blood cell folate assay |
| 3 | Awasthi S et al. | Serum folate concentration | <3 ng/mL | Chemiluminescent Microparticle Immunoassay (CMIA, Abbott Architect) |
| 4 | Bansal PG et al. | Serum folate concentration | <4 ng/mL | Chemiluminescence |
| 5 | Basiry M et al. | Serum folate concentration | <3 ng/ml | ELISA (Enzyme-Linked Immunosorbent Assay) |
| 6 | Daniel Jr WA et al. | Serum folate concentration | <2 ng/mL | Lactobacillus casei microbiological assay |
| 7 | de Lanerolle-Dias et al. | Serum folate concentration | <3 µg/L | Radioisotopic assay (SimulTRAC-SNBMP Biomedicals) |
| 8 | Demuyakor ME et al. | Serum folate concentration | < 7 nmol/L; | Microbiological assay (5MeTHF-specific) |
| 9 | Dhurde VS et al. | Serum folate concentration | <4 ng/mL | Electrochemiluminescence immunoassay (Roche Cobas e411) |
| 10 | EPHI | Serum folate concentration | <3 ng/ml | Not specified |
| 11 | Ercan S et al. | Serum folate concentration | <3 ng/ml | Chemiluminescent immunoassay (Roche Cobas e411) |
| 12 | Htet MK et al. | Serum folate concentration | <6.8 nmol/L | Microbiological assay (Lactobacillus rhamnosus) |
| 13 | Jani R. et al. | RBC folate concentration | <340 nmol/L | Radioimmunoassay (Dual Count Solid Phase No Boil) |
| 14 | Karakaş NM et al. | Serum folate concentration | <4 ng/mL | Chemiluminescence immunoassay |
| 15 | Kumar KJ et al. | Serum folate concentration | <2.7 ng/ml | ECLIA (Cobas E601E4) |
| 16 | Liebman M | RBC folate concentration | <140 ng/ml | Radioassay (125I kit, Becton Dickinson) |
| 17 | Monge-Rojas et al. | Serum folate concentration | <6.8 nmol/L | Solid Phase No Boil Dual 12 count kit (DPC) |
| 18 | Öner et al. | Serum folate concentration | <3 ng/ml | Chemiluminescent enzyme-labeled immunometric assay (Immulite kit) |
| 19 | Reiter LA et al. | Serum folate concentration | <3.0 ng/ml | Not specified |
| 20 | Saxena R et al. | RBC folate concentration | <317 nmol/L | Radioprotein-binding assay (SimulTRAC-S) |
| 21 | Shalini T et al. | RBC folate concentration | <151 ng/mL, | Chemiluminescence immunoassay (Siemens Centaur) |
| 22 | Thoradeniya T et al. | Serum folate concentration | <3 ng/mL | Competitive protein binding assay (SimulTRAC SNB radioassay) |
| 23 | Tsui JC et al. | Serum folate concentration | Not specified | immunoassay system |
| 24 | VanderJagt DJ et al. | Serum folate concentration | <6.8 nmol/L | Competitive protein-binding radioassay (SimulTRAC SNB) |
| 25 | Doshi et al. | RBC folate concentration | < 151 ng/mL | Not specified |
| 26 | Clark et al. | RBC folate concentration | < 140 ng/mL (317 nmol/L) | Microbiological assay with Lactobacillus casei |

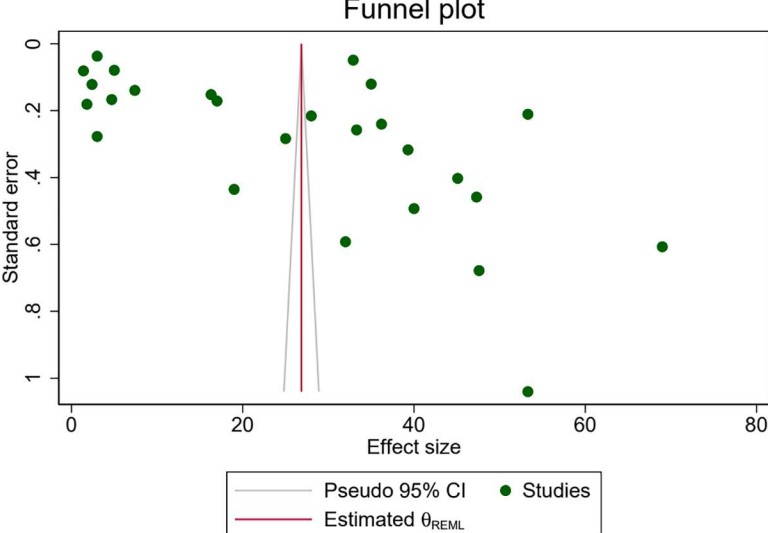

**Fig 7. Funnel plot for the study the proportion of folate defiecency among adolecent girls, 2025.**

**Table 4. Egger's of Publication Bias of Included Studies in the Systematic Review and Meta-Analysis the Proportion of Folate Defiecency Among Adolecent Girls, 2025.**

| Std_Eff | Coefficient | Std. err. | T | P>|t | [95% conf. interval] | |
|---------|-------------|-----------|------|-------|----------------------|---------|
| slope | 8.54 | 4.108764 | 2.08 | 0.05 | 0.0571956 | 17.01734 |
| bias | 60.33 | 35.2889 | 1.71 | 0.100 | −12.50358 | 133.1618 |

## 4. Discussion

The aim of this study was to determine the global prevalence of folate deficiency among adolescent girls. This is the first systematic review and meta-analysis conducted at a global level which determine the deficiency of folate among adolescent girls. The pooled prevalence of folate deficiency among adolescent girls was 26.9%. However, this estimate was derived from studies with substantial between-study heterogeneity and should therefore be interpreted as a highly variable summary measure across diverse populations and methodological contexts. On average, approximately one-quarter of adolescent girls were classified as folate deficient, although the true prevalence varied considerably across settings.. Folate is an essential micronutrient that plays a crucial role in adolescent health, supporting nervous system development, reproductive health, and the well-being of future offspring. However, despite its importance, a substantial proportion of adolescent girls worldwide continue to suffer from inadequate folate levels.

The prevalence of folate deficiency reported in the current study is lower than the highest prevalence rates previously observed in African countries. For instance a systematic review of African populations particularly women and children reported markedly high folate deficiency rates among women with prevalence estimates of 46.1% in Ethiopia, 79.2% in Sierra Leone, and 86.1% in Côte d'Ivoire [27]. The observed discrepancy may be attributed to the fact that women of reproductive age are more likely to experience folate deficiency than adolescent girls, largely due to increased physiological demands associated with menstruation and pregnancy, as well as a higher risk of dietary insufficiency [66]. In addition the higher prevalence of folate deficiency in Africa is largely driven by poor dietary patterns [67], technical and food system challenges for folic acid fortification [68]. Folate supplementation in Africa remains limited due to low service demand, inadequate funding, supply shortages, and weak program management [69]. Furthermore, low awareness of folate-rich

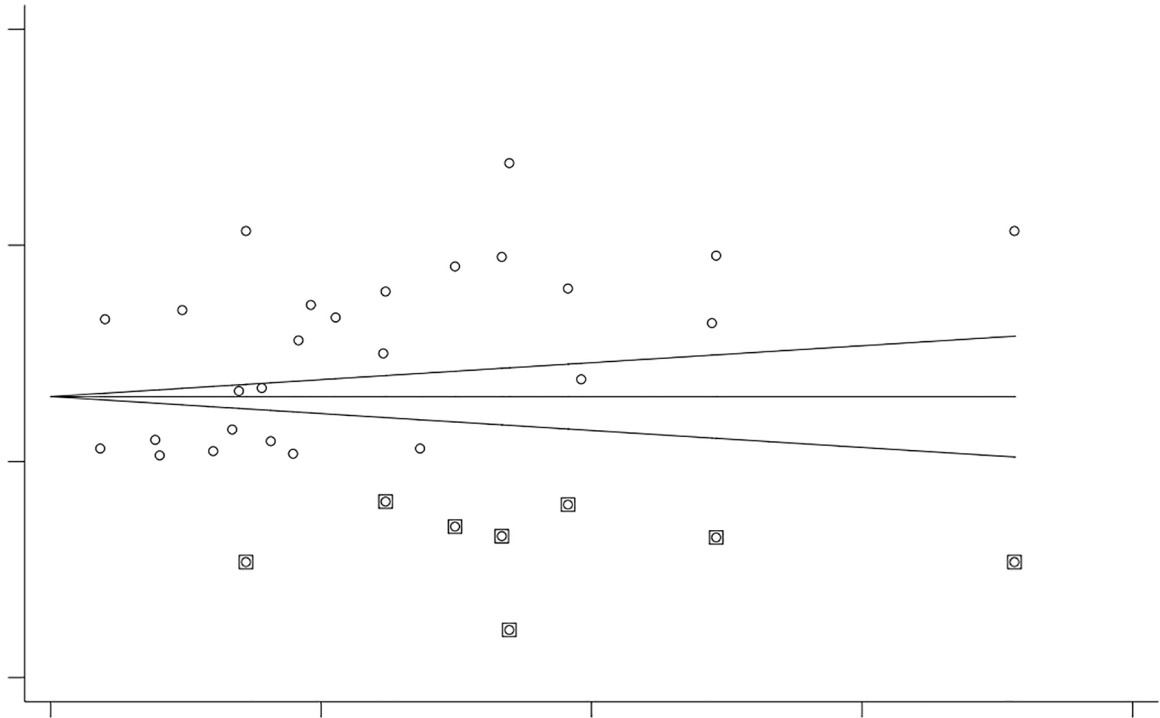

**Fig 8. Trim and fill analysis result of the study the proportion for folate defiecency among adolecent girls, 2025.**

foods, combined with socioeconomic barriers and restricted access to nutrient-dense foods, further increases the vulnerability of adolescents to folate deficiency. For instance study from Ethiopia report that over 80% of adolescent girls consume inadequate folate relative to dietary requirements [70].

On the other hand a systematic review from 39 countries reported that folate deficiency prevalence was typically less than 5% in high-income countries which is lower than the current study finding [66]. The discrepancy could be due to that high-income countries often have mandatory folic acid fortification programs and better access to folate-rich foods, leading to lower deficiency rates [71]. Additionally, these countries typically have more effective healthcare systems for infection prevention and treatment, further reducing the risk of folate deficiency.

In the subgroup analysis based on publication year, studies published between 2021 and 2024 reported the lowest prevalence at 21.9%. This lower prevalence could be attributed to several contributing factors. The main reason for the lower prevalence in recent studies is likely the implementation of interventions aimed at reducing folate deficiency among adolescent girls, including increased wide spread folate supplementation, mandatory food fortification of staple foods, and weekly iron–folate supplementation programs in schools [30,72]. For instance, research indicates that folate deficiency rates declined markedly after the fortification of wheat flour, dropping from approximately 7% before fortification to around 1.7% afterward [73]. Furthermore, improved dietary intake and public health interventions promoting the consumption of folate-rich foods have contributed to reducing folate deficiency among adolescent girls.

Based on subgroup analysis for folate deficiency assessment by specimen type, studies using RBC folate concentration reported a higher pooled prevalence (33.6%) compared to those measuring serum/plasma folate concentration (24.4%). This difference arises from the distinct biological properties of the two biomarkers. RBC folate reflects long-term folate status because it is incorporated into red blood cells during erythropoiesis and remains stable for the lifespan of the cell [74,75]. In contrast, serum/plasma folate is influenced by recent dietary intake and fluctuates with

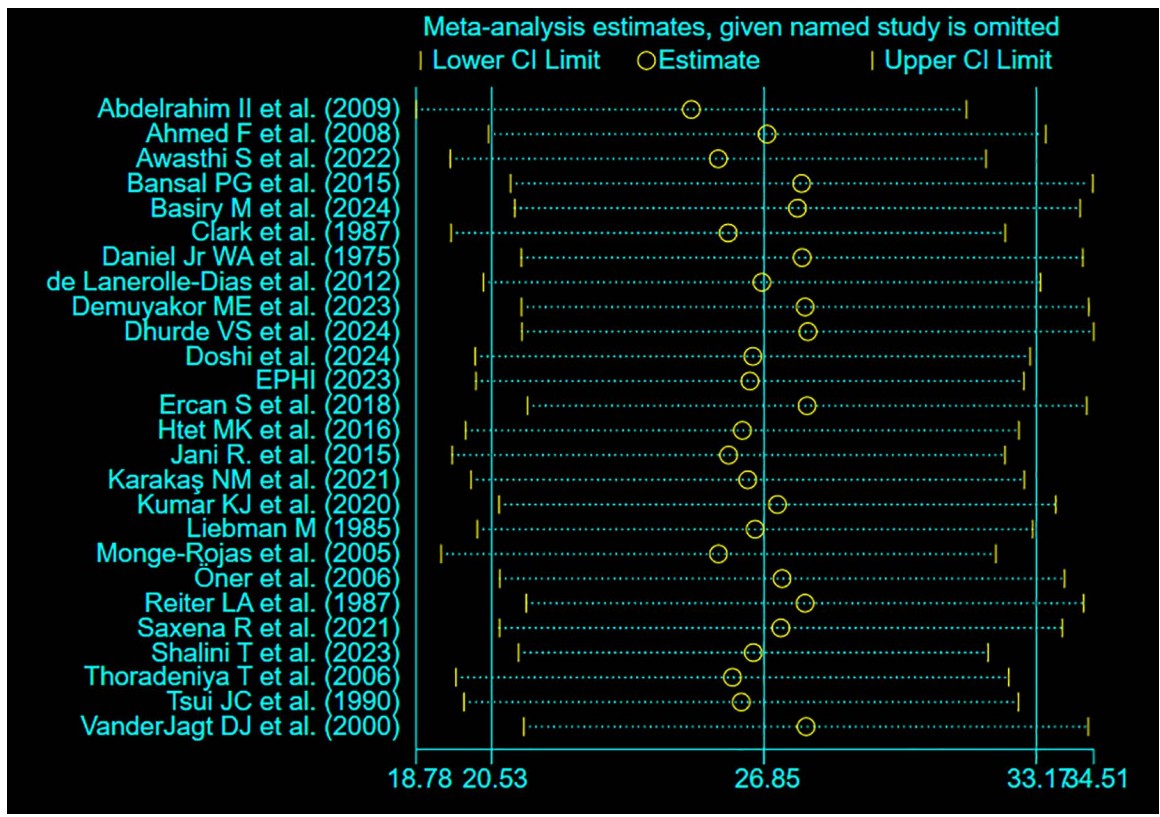

**Fig 9. Leave-one-out sensitivity analysis of pooled prevalence of folate deficiency among adolescent girls.**

**Table 5. Meta-regression analysis of study-level factors associated with heterogeneity in folate deficiency among adolescent girls.**

| Heterogeneity source | Coefficient | Std. err. | P-value |
|---|---|---|---|
| Publication year | −0.0338643 | 0.2113767 | 0.873 |
| Sample size | 0.0004282 | .0011545 | 0.711 |
| constant | 94.48272 | 424.5441 | 0.824 |

**Table 6. Certainty assessment for included outcomes for the folate defiecency among adolescent girls, 2025.**

| Outcome | Study design | Source | Risk of bias | Inconsistency | Directness | Imprecision | Publication bias | Proportion (CI) | Certainty |
|---|---|---|---|---|---|---|---|---|---|
| Folate Deficiency | Cross-sectional | 25 | Low[1] | Serious[2] | Direct[3] | low[4] | Very low[5] | 26.9% (20.5-33.2) | Low |
| | Longitudinal | 1 | low | Not applicabl | Direct | Low | not applicable | 47.6% | Low |

[1]Rated as low because most studies included were of high quality.

[2]Rated as Serious due to significant heterogeneity (I2 > 99.9%).

[3]Studies directly measure folate deficiency in adolescent girls

[4]Rated as low due to narrow confidence intervals.

[5]Publication bias was not present and rated as low.

short-term changes in folate consumption or supplementation [76]. Plasma/serum folate is strongly influenced by recent dietary intake and is less reliable for assessing tissue folate stores compared to RBC folate [76]. As a result, serum/plasma folate may underestimate the true burden of deficiency, while RBC folate provides a more reliable indicator of chronic folate depletion. This methodological distinction likely explains the higher prevalence observed in studies using RBC folate concentration.

Furthermore, the subgroup analysis revealed that the pooled prevalence of folate deficiency among adolescent girls in Africa was higher (35.5%) compared with other regions. This elevated prevalence may be explained by multiple factors, including limited dietary intake of folate-rich foods, cultural dietary restrictions, food insecurity, and inadequate nutrition education. Many adolescents in African countries consume monotonous diets dominated by staple cereals and tubers with limited intake of fruits, legumes, and green leafy vegetables, which are the main dietary sources of folate [77,78]. Food fortification programs are either absent or inconsistently implemented across several African countries. Poverty and food insecurity further exacerbate the problem, as families prioritize calorie-dense but micronutrient-poor foods over diverse diets. These factors collectively explain the disproportionately high prevalence of folate deficiency in African adolescents.

## 5. Strengths and limitations of the review

This systematic review and meta-analysis is the first to examine the global prevalence of folate deficiency among adolescents. A major strength of this review is its broad scope, which includes evidence from diverse regions and incorporates both published and unpublished studies, thereby reducing the risk of publication bias. Subgroup analyses were conducted to account for regional differences and minimize statistical heterogeneity.

Despite these strengths, several limitations must be acknowledged. The lack of studies from some countries limits the representativeness of the findings, which restricts their generalizability to all global populations. Considerable heterogeneity across studies, variations in assessment methods, and inconsistent cut-off points for folate deficiency pose challenges for direct comparison and synthesis. Another limitation of this review is that variations in laboratory assay methods and cut-off point definitions were not examined through subgroup or meta-regression analyses. These may have contributed substantially to the observed heterogeneity in prevalence estimates. We acknowledge that Egger's test may not be ideal for assessing publication bias in meta-analyses of proportions, particularly in the presence of substantial heterogeneity. Study quality was not incorporated into subgroup or meta-regression analyses which may limit interpretation of the pooled estimate.

## 6. Conclusions and recommendations

This review shows that folate deficiency is still a serious public health problem among adolescents around the world. Addressing this issue requires coordinated action at multiple levels. Efforts should focus on strengthening nutrition education, encouraging more diverse diets, and expanding food fortification and supplementation programs. It is also essential to improve access to folate-rich foods, especially in low- and middle-income countries where deficiencies are most common. Interventional studies are also necessary to identify the most effective and affordable strategies. Finally, updated and region-specific prevalence data will be critical for policymakers, health planners, and international organizations to design sustainable, evidence-based solutions for reducing folate deficiency in adolescents worldwide.

## Supporting information

**S1 File.** *Search terms for folate deficiency and its associated factors among adolescent girls: A systematic review and meta-analysis.*
(DOCX)

**S2 File. Data extraction table** *for folate deficiency and its associated factors among adolescent girls: A systematic review and meta-analysis.*
(XLSX)

**S3 File. Methodological quality assessment of included studies using Newcastle-Ottawa Scale (NOS).**
(DOCX)

**S4 File. PRISMA checklist.**
(DOCX)

## Acknowledgments

The authors would like to thank all authors of studies included in this systematic review and meta-analysis.

## Author contributions

**Conceptualization:** Mekuriaw Nibret Aweke, Nebebe Demis Baykemagn, Gebeyehu Lakew, Gebrie Getu Alemu, Berihun Agegn Mengistie.

**Data curation:** Nebebe Demis Baykemagn, Berihun Agegn Mengistie.

**Formal analysis:** Mekuriaw Nibret Aweke, Anas Ali Alhur, Berihun Agegn Mengistie.

**Investigation:** Astewil Moges Bazezew, Berihun Agegn Mengistie.

**Methodology:** Mekuriaw Nibret Aweke, Anas Ali Alhur, Nebebe Demis Baykemagn, Bisrat Tewelde Gebretsadkan, Berihun Agegn Mengistie.

**Project administration:** Bisrat Tewelde Gebretsadkan, Astewil Moges Bazezew, Amlaku Nigusie Yirsaw.

**Resources:** Mekuriaw Nibret Aweke, Berihun Agegn Mengistie.

**Software:** Gebrie Getu Alemu, Berihun Agegn Mengistie.

**Supervision:** Gebeyehu Lakew, Astewil Moges Bazezew, Wubet Tazeb Wondie, Berihun Agegn Mengistie.

**Visualization:** Mekuriaw Nibret Aweke, Gebeyehu Lakew, Gebrie Getu Alemu, Wubet Tazeb Wondie.

**Writing – original draft:** Mekuriaw Nibret Aweke, Nebebe Demis Baykemagn, Gebeyehu Lakew, Bisrat Tewelde Gebretsadkan, Gebrie Getu Alemu, Astewil Moges Bazezew, Amlaku Nigusie Yirsaw, Berihun Agegn Mengistie.

**Writing – review & editing:** Mekuriaw Nibret Aweke, Nebebe Demis Baykemagn, Gebeyehu Lakew, Bisrat Tewelde Gebretsadkan, Gebrie Getu Alemu, Astewil Moges Bazezew, Amlaku Nigusie Yirsaw, Wubet Tazeb Wondie, Berihun Agegn Mengistie.

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
