## [Decision Letter · Decision Letter 0]

15 Jan 2026

Dear Dr. Aweke,

plosone@plos.org. When you’re ready to submit your revision, log on to https://www.editorialmanager.com/pone/ and select the ‘Submissions Needing Revision’ folder to locate your manuscript file.. When you’re ready to submit your revision, log on to https://www.editorialmanager.com/pone/ and select the ‘Submissions Needing Revision’ folder to locate your manuscript file.. When you’re ready to submit your revision, log on to https://www.editorialmanager.com/pone/ and select the ‘Submissions Needing Revision’ folder to locate your manuscript file.. When you’re ready to submit your revision, log on to https://www.editorialmanager.com/pone/ and select the ‘Submissions Needing Revision’ folder to locate your manuscript file.

We look forward to receiving your revised manuscript.

Kind regards,

Marly A. Cardoso, Ph.D.

Academic Editor

PLOS One

Journal Requirements:

2. We note you have included a table to which you do not refer in the text of your manuscript. Please ensure that you refer to Table 4 in your text; if accepted, production will need this reference to link the reader to the table.

3. Please include captions for your Supporting Information files at the end of your manuscript, and update any in-text citations to match accordingly. Please see our Supporting Information guidelines for more information: http://journals.plos.org/plosone/s/supporting-information....

Reviewers' comments:

Reviewer's Responses to Questions

**Comments to the Author**

1. Is the manuscript technically sound, and do the data support the conclusions?

Reviewer #1: Yes

Reviewer #2: Partly

2. Has the statistical analysis been performed appropriately and rigorously?

Reviewer #1: Yes

Reviewer #2: Yes

3. Have the authors made all data underlying the findings in their manuscript fully available?

Reviewer #1: Yes

Reviewer #2: Yes

4. Is the manuscript presented in an intelligible fashion and written in standard English?

Reviewer #1: Yes

Reviewer #2: Yes

Reviewer #1: Relevant and well-written article. Just a few points to be reviewed:

1. Abstract

-The results do not indicate how many studies were captured in the search strategy and how many were included, which is necessary according to the PRISMA-2020 checklist guidelines for abstracts.

-In conclusions, the excerpt "Folate deficiency among adolescent girls is highly prevalent globally, affecting more than one in four adolescent girls. The burden is particularly higher in Africa, where over one-third of adolescent girls are affected." repeats exactly what was said in the results. I recommend removing it.

2. Methods

-The definition of adolescence can vary culturally. Indicate which age range was considered to define this life stage.

-In Supplementary File 1 (S1), the only table inserted is titled S3.

-The inclusion and exclusion criteria lack the use of an acronym (such as PECOT) in the form of a table to more clearly indicate all the criteria.

-To assess the degree of agreement between reviewers, it would be useful to calculate Cohen's Kappa index.

- In the assessment of the quality of the individual studies, it is stated that a modified version of the Newcastle-Ottawa Scale for cross-sectional studies was used. What modification? Is it validated? Was it modified to evaluate non-cross-sectional studies? Why not use other tools specifically designed for this purpose?

-In data extraction, it would be interesting to determine if the studies have conflicts of interest.

3. Results

-In Table 1:

a. The studies are ordered alphabetically. This is not incorrect, but it is generally done by the year of publication of the studies, establishing a chronology that facilitates understanding the evolution of the research.

b. There is a lack of standardization in the table, with no initial capital letter for "Mean of age" and "Type of study" in the last two included studies.

c. It is important to indicate the reference number in the first column or in parentheses next to the authors' names.

-In Table 5 of the GRADE assessment, although almost all studies were cross-sectional, separating them by design, rather than simply grouping them all as observational, enriches the reader's understanding.

Reviewer #2: General Comments

The study has several strengths, including adherence to the PRISMA guidelines, prospective registration in PROSPERO, and the availability of complete search strategies for each database. These aspects contribute positively to transparency and methodological rigor. However, some methodological aspects still require clarification or stronger justification to ensure reproducibility and alignment with best practices for prevalence meta-analyses.

Abstract

• The abstract does not provide sufficient methodological detail for readers to properly assess the robustness of the review.

• Minor grammatical and stylistic revisions are needed to improve clarity and readability.

• The objective statement contains a grammatical error (“aims to addresses”) that should be corrected.

• Percentages should be formatted consistently, with appropriate spacing and punctuation (e.g., “26.9% (95% CI: 20.5–33.2)”).

Introduction

• The objective of the study is not clearly stated at the end of the Introduction. The final paragraph should explicitly present the study aim to ensure coherence with the Methods and Results sections.

Methods

The Methods section is generally well organized and follows the standard structure expected for a systematic review and meta-analysis. The authors report protocol registration, adherence to PRISMA guidelines, duplicate screening, independent data extraction, and the use of established tools for quality and certainty assessment. Overall, this reflects an adequate methodological framework.

However, several points require clarification or better justification to improve transparency and reproducibility. Many of the elements listed below appear later in the Results section, suggesting that they were considered during the review process. Nonetheless, these aspects should be clearly described in the Methods section, rather than being left implicit, to avoid ambiguity for the reader. I therefore suggest revising the Methods to improve organization and clarity.

• The study population is not clearly defined. The authors should explicitly state the age range used to define “adolescents” and clarify whether only healthy adolescent girls were included or whether those with underlying health conditions were also eligible.

• It is unclear whether the final version of the review strictly followed the registered PROSPERO protocol. The authors should clearly state whether any deviations occurred after registration (e.g., changes in eligibility criteria, outcomes, or subgroup analyses), and, if so, provide justification.

• The rationale for the choice of databases should be better explained. For this topic, other widely used databases such as CINAHL, the Cochrane Library, PsycINFO, and Scopus are commonly included. In addition, the inclusion of HINARI and DOAJ alongside PubMed and ScienceDirect should be justified, given the potential overlap in indexed content.

• The inclusion of Google and Google Scholar as primary data sources raises concerns about reproducibility. Although these platforms are sometimes used to identify grey literature, the Methods section does not describe how searches were standardized (e.g., number of pages screened, sorting criteria, or limits applied).

• It is not clear whether controlled vocabulary terms (e.g., MeSH terms) were used in PubMed or whether the search relied exclusively on free-text terms. This information is important to assess the sensitivity and completeness of the search strategy.

• The Methods section would benefit from explicitly stating the review question (e.g., using a PECO or similar framework).

• Although studies reporting folate deficiency were included, acceptable biomarkers (e.g., serum folate versus red blood cell folate) and diagnostic cut-off values are not specified. Given the large variability in cut-offs across settings and over time, this omission is important and likely contributes substantially to the extreme heterogeneity observed. While laboratory methods and cut-offs are listed as extracted variables, it remains unclear how this variability was handled analytically (e.g., subgroup analyses, sensitivity analyses, or standardization).

• The inclusion of both published and unpublished studies increases comprehensiveness, but also raises concerns regarding study quality and selective availability, which should be addressed more explicitly.

• The Methods section does not specify which version or adaptation of the Newcastle–Ottawa Scale was used, nor whether the adapted version has been validated for cross-sectional studies.

• The justification for using a random-effects model is generic. Given the expected heterogeneity in prevalence studies, the authors should specify which estimator was used (e.g., DerSimonian–Laird or restricted maximum likelihood [REML]).

• The significance level used for funnel plots and publication bias assessment should be clearly stated.

• The authors do not report whether prevalence estimates were transformed (e.g., logit transformation) prior to pooling, which is an important methodological consideration in meta-analyses of proportions.

• The use of Egger’s test to assess publication bias is questionable in this context, as it is not well suited for proportion data, particularly in the presence of extreme heterogeneity. This limitation should be acknowledged or alternative approaches considered.

Results

• The statement “Most studies were of moderate to high quality, with scores ranging from five to eight” is imprecise. The authors should report the exact number and proportion of studies in each quality category.

• Figure 1 is appropriately referenced; however, minor typographical errors should be corrected (e.g., “Defiecency,” “adolecent”).

• The statement that a fixed-effect model “indicated significant heterogeneity” is conceptually incorrect. Heterogeneity is assessed independently of the choice of meta-analytic model. This sentence should be revised for conceptual accuracy.

• Given the extremely high heterogeneity, the Results section should more clearly emphasize that the pooled estimate represents an average across highly heterogeneous contexts, rather than a precise global prevalence.

• The interpretation of the GRADE assessment is somewhat inconsistent. Although the certainty of evidence is rated as low due to heterogeneity, the Results section simultaneously states that “the majority of studies were of high quality,” which may be confusing. It should be clarified that high methodological quality of individual studies does not compensate for inconsistency and indirectness at the body-of-evidence level.

Funding

The Funding section was left empty.

.

Reviewer #1: No

Reviewer #2: **Yes:** Ana Carolina de Andrade HovadickAna Carolina de Andrade HovadickAna Carolina de Andrade HovadickAna Carolina de Andrade Hovadick

---

## [Author Response · Author response to Decision Letter 1]

5 Feb 2026

29 Jan, 2026

Dear Editors of PLOS ONE,

We thank you and the reviewers for the time, effort, and expertise dedicated to the evaluation of our manuscript, PONE-D-25-60905 , entitled “Global Folate Deficiency Among Adolescent Girls: A Systematic Review and Meta-analysis.” We greatly appreciate the constructive and insightful comments provided, which have been invaluable in improving the clarity, rigor, and overall quality of the manuscript.

In response to the reviewers’ feedback, we have carefully revised the manuscript and prepared a detailed, point-by-point response explaining how each comment has been addressed. All modifications have been clearly indicated in the revised manuscript to facilitate review. We believe that these revisions have strengthened the analysis, interpretation, and presentation of our findings.

Thank you for the opportunity to revise and resubmit our work. We appreciate your consideration and look forward to your further assessment.

Sincerely

Mekuriaw Nibret Aweke

Corresponding Author

1. Editors comment

The overall topic is of value in Public Health Nutrition. However, there are many important methodological aspects for a major review as pointed out by the reviewers. A special attention should be provided for the following recommendations: • The inclusion of Google and Google Scholar as primary data sources raises concerns about reproducibility; • It is not clear whether controlled vocabulary terms (e.g., MeSH terms) were used in PubMed or whether the search relied exclusively on free-text terms; and • The Methods section would benefit from explicitly stating the review question (e.g., using a PECO or similar framework). Please submit your revised manuscript

Authers response:

Thank you for the opportunity to revise our manuscript in response to the editors’ and reviewers’ comments and suggestions. We have carefully revised the manuscript, with particular attention to strengthening the search strategy through refinement of the search terms (including MeSH terms) and by explicitly stating the review question using the PICO framework. We believe these revisions have improved the methodological clarity and robustness of the manuscript.

Authers response:

Dear editor, thank you very much for the reminder to meets the PLOS ONE’S style requirement. We have revised the manuscript in accordance with the journals style requirement including file naming.

2. We note you have included a table to which you do not refer in the text of your manuscript. Please ensure that you refer to Table 4 in your text; if accepted, production will need this reference to link the reader to the table.

Authers response:

Dear editor, thank you very much for your request to refer table 4 in the text. We noted that we didn’t refer table 4 and now we have revised it and we ensure that all the table in the manuscript were correctly referred. Thank you for your request.

3. Please include captions for your Supporting Information files at the end of your manuscript, and update any in-text citations to match accordingly.

Authers response:

Dear revewer, thank you very much you’re your request to add supporting information files at the end of our manuscript. We have added the supporting information files at the end of our manuscript.

1. Revewer #1 comments

Relevant and well-written article. Just a few points to be reviewed:

Response to Reviewer:

Thank you for your positive feedback on the relevance and importance of our study. We value your comments and suggestions, which have helped us improve the manuscript.

1. Abstract

The results do not indicate how many studies were captured in the search strategy and how many were included, which is necessary according to the PRISMA-2020 checklist guidelines for abstracts.

-In conclusions, the excerpt "Folate deficiency among adolescent girls is highly prevalent globally, affecting more than one in four adolescent girls. The burden is particularly higher in Africa, where over one-third of adolescent girls are affected." repeats exactly what was said in the results. I recommend removing it.

Response to Reviewer:

Dear Reviewer, Thank you for this comment. We have revised the abstract to explicitly report the number of records identified and the number of studies included, in accordance with the PRISMA-2020 checklist for abstracts. The revise statement reads as the following:

“The search strategy identified 1,498 records, of which 26 studies met the eligibility criteria and were included in this systematic review and meta-analsyis.”

In addition in the conclusion section we had removed the repeated sttatment which will enhance the readability of the abstract sections. Thank you very much for this crucial and insightful suggesions.

2. Methods

-The definition of adolescence can vary culturally. Indicate which age range was considered to define this life stage.

Response to Reviewer:

Dear Reviewer,

Thank you for this comment. We have clarified the definition of adolescence in the Methods section. Adolescence was defined as ages 10–19 years, consistent with the World Health Organization classification, and this age range was applied as an inclusion criterion for the review.

-In Supplementary File 1 (S1), the only table inserted is titled S3.

Response to Reviewer:

Dear Reviewer,

Thank you for pointing this inconsistent labeling of supplmentary files. The supplementary files were correctly labeled as Supplementary Files 1–4. We have corrected the table labeling so that each supplementary file now contains a consistently numbered table corresponding to its file number, and all in-text references have been updated accordingly.

-The inclusion and exclusion criteria lack the use of an acronym (such as PECOT) in the form of a table to more clearly indicate all the criteria.

Response to Reviewer:

Dear Reviewer,

Thank you for suggesting the inclusion of the PECOT framework for our eligibility criteria. We have now added a table presenting the PECOT criteria used in our study. We hope this provides a clear and concise overview of the inclusion and exclusion criteria, and we greatly appreciate your insightful recommendation.

-To assess the degree of agreement between reviewers, it would be useful to calculate Cohen's Kappa index.

Response to Reviewer:

Dear Reviewer, we sincerely appreciate your suggestion regarding the calculation of Cohen’s Kappa. While we did not calculate the statistic, all discrepancies between reviewers during study selection were carefully discussed and resolved by consensus, ensuring accurate and consistent inclusion and exclusion decisions.

- In the assessment of the quality of the individual studies, it is stated that a modified version of the Newcastle-Ottawa Scale for cross-sectional studies was used. What modification? Is it validated? Was it modified to evaluate non-cross-sectional studies? Why not use other tools specifically designed for this purpose?

Response to Reviewer:

We thank the reviewer for this important point. We used a modified version of the Newcastle–Ottawa Scale (NOS) adapted for cross-sectional studies, following published guidance in the literature. The modifications involved adapting the selection and outcome assessment items to be applicable to cross-sectional designs, while retaining the core scoring system for representativeness, sample size, comparability, and outcome assessment.

This modified tool has been widely used and cited in systematic reviews of prevalence studies. It was not applied to non-cross-sectional studies. We selected this tool because it allows structured and reproducible assessment of methodological quality in observational studies.

-In data extraction, it would be interesting to determine if the studies have conflicts of interest.

Response to Reviewer:

We thank the reviewer for the suggestion to report potential conflicts of interest. We carefully reviewed each included study and did not identify any reports of conflicts of interest.

3. Results

-In Table 1:

a. The studies are ordered alphabetically. This is not incorrect, but it is generally done by the year of publication of the studies, establishing a chronology that facilitates understanding the evolution of the research.

Response to Reviewer:

We thank the reviewer for valuable recommendation to order the studies based on year of publication. We have revised the table and studies are ordered based on the publication year and we hope new it becomes easier to understand the evolution of the research. Thank you very much for this crucial comment.

b. There is a lack of standardization in the table, with no initial capital letter for "Mean of age" and "Type of study" in the last two included studies.

Response to Reviewer:

We thank the reviewer for the valuable recommendation to standardize the table by correcting the initial capital letters for "Mean of Age" and "Type of Study." We have now standardized the table by capitalizing the initial letter of each phrase or word in all table columns and rows. We hope this improves the table’s consistency and readability.

c. It is important to indicate the reference number in the first column or in parentheses next to the authors' names.

Response to Reviewer:

We thank the reviewer for this valuable suggestion. In response, we have updated the table to include references using the “Author (Reference)” format as recommended.

-In Table 5 of the GRADE assessment, although almost all studies were cross-sectional, separating them by design, rather than simply grouping them all as observational, enriches the reader's understanding.

Response to Reviewer:

We thank the reviewer for this valuable suggestion. We acknowledge that although the majority of our included studies were cross-sectional (25 out of 26), separating them by study design rather than grouping all as “observational” provides clearer information. In response, we have updated Table 5 of the GRADE assessment to present the cross-sectional studies and the single longitudinal l study separately, allowing a more precise understanding of the evidence quality for each design.

Reviewer #2: General Comments

The study has several strengths, including adherence to the PRISMA guidelines, prospective registration in PROSPERO, and the availability of complete search strategies for each database. These aspects contribute positively to transparency and methodological rigor. However, some methodological aspects still require clarification or stronger justification to ensure reproducibility and alignment with best practices for prevalence meta-analyses.

Response to Reviewer:

We appreciate the positive feedback and explanation about the methodological rigor. We have revised the document in each section including methodological clarification by including stronger justification to ensure reproducibility and alignments. We hope this will enhance the quality of the work.

Abstract

• The abstract does not provide sufficient methodological detail for readers to properly assess the robustness of the review.

Response to Reviewer:

We thank the reviewer for this insightful comment. In response, we have revised the abstract to include additional methodological details to enhance transparency and allow readers to better assess the robustness of the review. Specifically, the abstract methods section now states:

“Publication bias was evaluated through visual inspection of funnel plots and Egger’s regression test, and adjusted estimates were calculated using the trim-and-fill method. Meta-regression analyses were conducted to explore potential sources of heterogeneity. Sensitivity analyses were performed to assess the robustness of pooled estimates.”

• Minor grammatical and stylistic revisions are needed to improve clarity and readability.

• The objective statement contains a grammatical error (“aims to addresses”) that should be corrected.

Response to Reviewer

We thank the reviewer for this valuable comment. The grammatical and typographical errors in the abstract have been corrected, and we believe that the revisions have improved the clarity and overall quality of the manuscript.

• Percentages should be formatted consistently, with appropriate spacing and punctuation (e.g., “26.9% (95% CI: 20.5–33.2)”).

Response to Reviewer

We thank the reviewer for this very valuable suggesions to use consistent writing format to report the pooled prevalence. Based on the suggesions we have correct the inconsistent punctuation and writing format throughout the manuscript and now we used “26.9% (95% CI: 20.5–33.2) through out the document. Thank you very much for this important comment.

Introduction

• The objective of the study is not clearly stated at the end of the Introduction. The final paragraph should explicitly present the study aim to ensure coherence with the Methods and Results sections.

Response to Reviewer:

Dear reviewer thank you very much for this insightful comment to add the study objective in the introduction section. Accordingly we had added the clear and precise of this study objective at the end of the introduction section and the final sttement read as the following:

“The objective of this study was to systematically review and quantitatively synthesize the available evidence to estimate the pooled prevalence of folate deficiency among adolescent girls and to inform targeted public health interventions.”

Methods

The Methods section is generally well organized and follows the standard structure expected for a systematic review and meta-analysis. The authors report protocol registration, adherence to PRISMA guidelines, duplicate screening, independent data extraction, and the use of established tools for quality and certainty assessment. Overall, this reflects an adequate methodological framework. However, several points require clarification or better justification to improve transparency and reproducibility.

Response to Reviewer:

Dear reviewer, we are glad to hear this acknowledgment and consideration about the adequate methodological framework. Thank you again and we hope all other issues are adressed in the revised manusrcipt.

Many of the elements listed below appear later in the Results section, suggesting that they were considered during the review process. Nonetheless, these aspects should be clearly described in the Methods section, rather than being left implicit, to avoid ambiguity for the reader. I therefore suggest revising the Methods to improve organization and clarity.

Response to Reviewer:

Dear Reviewer, we sincerely appreciate your valuable suggestions and constructive comments regarding the Methods section. We have carefully revised the Methods section to address each of the raised concerns and clarify aspects that were previously insufficiently described. We hope that these revisions have adequately addressed the issues highlighted.

The study population is not clearly defined. The authors should explicitly state the age range used to define “adolescents” and clarify whether only healthy adolescent girls were included or whether those with underlying health conditions were also eligible.

Response to Reviewer:

Dear Reviewer, we thank you for raising these important concerns regarding clarification of the adolescent age range and the inclusion criteria related to health status. We have revised the Methods section to improve clarity by explicitly specifying that adolescent girls aged 10–19 years were included in the study and that only apparently healthy adolescent girls were considered. These details are now clearly stated in the Methods section of the revised manuscript.

• It is unclear whether the final version of the review strictly followed the registered PROSPERO protocol. The authors should clearly state whether any deviations occurred after registration (e.g., changes in eligibility criteria, outcomes, or subgroup analyses), and, if so, provide justification.

Response to Reviewer:

We thank the reviewer for this important comment. We confirm that the final version of the review strictly adhered to the registered PROSPERO protocol. No deviations w

---

## [Decision Letter · Decision Letter 1]

3 Mar 2026

Dear Dr. Aweke,

We look forward to receiving your revised manuscript.

Kind regards,

Marly A. Cardoso, Ph.D.

Academic Editor

PLOS One

Journal Requirements:

Reviewers' comments:

Reviewer's Responses to Questions

**Comments to the Author**

Reviewer #1: (No Response)

Reviewer #2: All comments have been addressed

2. Is the manuscript technically sound, and do the data support the conclusions?

Reviewer #1: Yes

Reviewer #2: Yes

3. Has the statistical analysis been performed appropriately and rigorously?

Reviewer #1: Yes

Reviewer #2: Yes

4. Have the authors made all data underlying the findings in their manuscript fully available?

Reviewer #1: Yes

Reviewer #2: Yes

5. Is the manuscript presented in an intelligible fashion and written in standard English?

Reviewer #1: Yes

Reviewer #2: Yes

Reviewer #1: This manuscript addresses an important global public health issue and is generally well structured. The authors have adequately addressed several points raised in the previous round of review, including clarification of methodological procedures, improvements in the presentation of results, and refinement of certain analytical descriptions.

Despite these improvements, some methodological and interpretative issues remain:

1. Extreme heterogeneity (I² > 99%)

The level of heterogeneity is extremely high. Although subgroup analyses and sensitivity analyses have been performed, the pooled estimate (26.9%) continues to be interpreted in relatively strong terms (e.g., “The findings revealed that 26.9% of adolescent girls are affected by folate deficiency. Approximately one in every four adolescent girls suffers from folate deficiency.”).

Given the magnitude of heterogeneity, the manuscript would benefit from a clearer justification for pooling under these conditions and a more cautious interpretation of the overall prevalence estimate. The pooled result should be framed explicitly as a highly heterogeneous summary measure across diverse populations and methodological contexts.

2. Biomarkers, Laboratory Methods, and Cut-off Points

The manuscript does not sufficiently explore whether differences in biomarker type (e.g., serum folate versus erythrocyte folate), laboratory methods, or cut-off point definitions were analytically examined as potential sources of heterogeneity. These methodological differences are highly relevant in micronutrient deficiency research and can substantially influence prevalence estimates.

Subgroup analyses or meta-regression incorporating these variables would strengthen the conclusions. If such analyses are not feasible, this limitation should be discussed more explicitly, particularly as a potential contributor to the observed heterogeneity.

3. Integration of Risk of Bias into the Quantitative Analysis

Although a modified version of the Newcastle–Ottawa Scale was applied and adequately described, the analysis does not assess whether study quality influenced the pooled estimate. Incorporating study quality into subgroup analyses or meta-regression, or explicitly discussing the limitations of not integrating quality into the quantitative synthesis, would improve interpretability.

Reviewer #2: The authors have adequately addressed the previous comments and revised the manuscript in line with expectations. However, a conceptual issue remains in the PECOT framework. Although the search strategy is appropriate, the PECOT is not fully aligned with it. Specifically, the exposure (E) is not appropriately defined. Folate deficiency represents the outcome of interest rather than an exposure, and listing it as such may be conceptually misleading. In prevalence studies and systematic reviews focused on nutritional status, the exposure typically refers to risk factors associated with the condition, or it may be designated as not applicable (N/A) when the primary objective is purely descriptive. For greater conceptual clarity and methodological consistency, it would be preferable to redefine the exposure as “risk factors for folate deficiency” or to indicate it as N/A.

.

Reviewer #1: No

Reviewer #2: **Yes:** Ana Carolina HovadickAna Carolina HovadickAna Carolina HovadickAna Carolina Hovadick

---

## [Author Response · Author response to Decision Letter 2]

4 Mar 2026

04 March, 2026

Dear Editors of PLOS ONE,

We thank you and the reviewers for the time, effort, and expertise dedicated to the evaluation of our manuscript, PONE-D-25-60905 , entitled “Global Folate Deficiency Among Adolescent Girls: A Systematic Review and Meta-analysis.” We greatly appreciate the constructive and insightful comments provided, which have been invaluable in improving the clarity, rigor, and overall quality of the manuscript.

In response to the reviewers’ feedback, we have carefully revised the manuscript and prepared a detailed, point-by-point response explaining how each comment has been addressed. All modifications have been clearly indicated in the revised manuscript to facilitate review. We believe that these revisions have strengthened the analysis, interpretation, and presentation of our findings.

Thank you for the opportunity to revise and resubmit our work. We appreciate your consideration and look forward to your further assessment.

Sincerely

Mekuriaw Nibret Aweke

Corresponding Author

1. Editors comment

Thank you for submitting your manuscript to PLOS ONE. After careful consideration, we feel that it has merit but does not fully meet PLOS ONE’s publication criteria as it currently stands. Therefore, we invite you to submit a revised version of the manuscript that addresses the points raised during the review process.

Authers response:

Thank you for the opportunity to revise our manuscript in response to the editors’ and reviewers’ comments and suggestions. We have revised the manuscript based on the reviewers comment and submitted accordingly. We highly appreciate the editors continues follow up and opportunity to revise the work for better enhancement of the quality of the work..

1. Revewer #1 comments

This manuscript addresses an important global public health issue and is generally well structured. The authors have adequately addressed several points raised in the previous round of review, including clarification of methodological procedures, improvements in the presentation of results, and refinement of certain analytical descriptions.

Response to Reviewer:

Dear Reviewer, thank you very much for the positive feedback and acknowledgment of the addressing several points raised before. We also highly appreciate the reviewer for the continues reviewing and detail comments and suggesions that improve the quality of the work.

1. Extreme heterogeneity (I² > 99%)

The level of heterogeneity is extremely high. Although subgroup analyses and sensitivity analyses have been performed, the pooled estimate (26.9%) continues to be interpreted in relatively strong terms (e.g., “The findings revealed that 26.9% of adolescent girls are affected by folate deficiency. Approximately one in every four adolescent girls suffers from folate deficiency.”). Given the magnitude of heterogeneity, the manuscript would benefit from a clearer justification for pooling under these conditions and a more cautious interpretation of the overall prevalence estimate. The pooled result should be framed explicitly as a highly heterogeneous summary measure across diverse populations and methodological contexts.

Response to Reviewer:

Dear Reviewer,

Thank you for your suggestion to consider the higher hetrogeniety between studys and interpret the result by considering this higher hetrogeniety.

We have revised the above statement and corrected as the following

“The pooled prevalence of folate deficiency among adolescent girls was 26.9%. However, this estimate was derived from studies with substantial between-study heterogeneity and should therefore be interpreted as a highly variable summary measure across diverse populations and methodological contexts. On average, approximately one-quarter of adolescent girls were classified as folate deficient, although the true prevalence varied considerably across settings.”

We hope this revision makes the interpretation of the finding more appropraite and consider the heterogeneity.

2. Biomarkers, Laboratory Methods, and Cut-off Points

The manuscript does not sufficiently explore whether differences in biomarker type (e.g., serum folate versus erythrocyte folate), laboratory methods, or cut-off point definitions were analytically examined as potential sources of heterogeneity. These methodological differences are highly relevant in micronutrient deficiency research and can substantially influence prevalence estimates.

Subgroup analyses or meta-regression incorporating these variables would strengthen the conclusions. If such analyses are not feasible, this limitation should be discussed more explicitly, particularly as a potential contributor to the observed heterogeneity.

Respon0se to Reviewer:

Dear Reviewer,

Thank you for suggesting the inclusion of the PECOT framework for our eligibility criteria. We have now added a table presenting the PECOT criteria used in our study. We hope this provides a clear and concise overview of the inclusion and exclusion criteria, and we greatly appreciate your insightful recommendation.

-To assess the degree of agreement between reviewers, it would be useful to calculate Cohen's Kappa index.

Response to Reviewer:

Dear Reviewer, we sincerely appreciate your valuable suggestion regarding variations in study cut-off points and laboratory assay methods. Although we conducted a subgroup analysis based on serum type, we were unable to perform subgroup or meta-regression analyses according to cut-off definitions and laboratory assay methods. We have now revised the manuscript to explicitly acknowledge this as a limitation as the following:

“Another limitation of this review is that variations in laboratory assay methods and cut-off point definitions were not examined through subgroup or meta-regression analyses. These may have contributed substantially to the observed heterogeneity in prevalence estimates”

3. Integration of Risk of Bias into the Quantitative Analysis

Although a modified version of the Newcastle–Ottawa Scale was applied and adequately described, the analysis does not assess whether study quality influenced the pooled estimate. Incorporating study quality into subgroup analyses or meta-regression, or explicitly discussing the limitations of not integrating quality into the quantitative synthesis, would improve interpretability.

Response to Reviewer:

Dear Reviewer, we thank you for your constructive comment regarding the integration of study quality into the quantitative synthesis. Although study quality was assessed using a modified version of the Newcastle–Ottawa Scale, we did not incorporate risk-of-bias scores into subgroup or meta-regression analyses. Due to limited variability in quality ratings and incomplete reporting across some studies, such analyses were not feasible. We have now explicitly acknowledged this as a limitation in the revised manuscript to improve interpretability of the pooled estimate as the following:

“Study quality was not incorporated into subgroup or meta-regression analyses, which may limit interpretation of the pooled estimate.”

Reviewer #2: General Comments

The authors have adequately addressed the previous comments and revised the manuscript in line with expectations. However, a conceptual issue remains in the PECOT framework. Although the search strategy is appropriate, the PECOT is not fully aligned with it. Specifically, the exposure (E) is not appropriately defined. Folate deficiency represents the outcome of interest rather than an exposure, and listing it as such may be conceptually misleading. In prevalence studies and systematic reviews focused on nutritional status, the exposure typically refers to risk factors associated with the condition, or it may be designated as not applicable (N/A) when the primary objective is purely descriptive. For greater conceptual clarity and methodological consistency, it would be preferable to redefine the exposure as “risk factors for folate deficiency” or to indicate it as N/A.

Response to Reviewer:

We sincerely appreciate the positive feedback. We acknowledge that the exposure was not clearly defined, as folate deficiency represents the outcome rather than the exposure. The manuscript has been revised accordingly, and the exposure is now indicated as “Not applicable.”

We are also grateful for the reviewer’s detailed, section-specific comments, which have been invaluable in strengthening the overall quality of the manuscript.

---

## [Editor Report · Decision Letter 2]

22 Mar 2026

Global Folate Deficiency Among Adolescent Girls: A Systematic Review and Meta-analysis

PONE-D-25-60905R2

Dear Dr. Aweke,

We’re pleased to inform you that your manuscript has been judged scientifically suitable for publication and will be formally accepted for publication once it meets all outstanding technical requirements.

Kind regards,

Marly A. Cardoso, Ph.D.

Academic Editor

PLOS One

---

## [Editor Report · Acceptance letter]

PONE-D-25-60905R2

PLOS One

Dear Dr. Aweke,

I'm pleased to inform you that your manuscript has been deemed suitable for publication in PLOS One. Congratulations! Your manuscript is now being handed over to our production team.

Kind regards,

on behalf of

Dr. Marly A. Cardoso

Academic Editor

PLOS One